# AN EFFICIENT MEAN-FIELD APPROACH TO HIGH-ORDER MARKOV LOGIC

## ABSTRACT

Markov logic networks (MLNs) are powerful models for symbolic reasoning, which combine probabilistic modeling with relational logic. Inference algorithms for MLNs often perform at the level of propositional logic or require building a first-order probabilistic graph, and the computational efficiency remains a challenge. The mean-field algorithm generalizes message passing for approximate inference in many intractable probabilistic graphical models, but in MLNs it still suffers from the high-order dependencies among the massive groundings, resulting in time complexity exponential in both the length and the arity of logic rules. We propose a novel method, LogicMP, to simplify the logic message passing especially. In most practical cases, it can reduce the complexity significantly to polynomial for the formulae in conjunctive normal form (CNF). We exploit the property of CNF logic rules to sidestep the expectation computation of high-order dependency, and then formulate the logic message passing by Einstein summation to facilitate parallel computation, which can be optimized by sequentially contracting the rule arguments. With LogicMP, we achieve evident improvements on several reasoning benchmark datasets in both performance and efficiency over competitor methods. Specifically, the AUC-PR of the UW-CSE and Cora datasets is improved by more than 11% absolutely and the speed is about ten times faster.

## 1 INTRODUCTION

Despite the remarkable improvement in deep learning, the ability of symbolic learning is believed to be indispensable for the development of modern AI (Besold et al., 2021). The entities in the real world are interconnected with each other through various relationships, forming massive relational data, which leads to many logic-based models (Koller et al., 2007). Markov logic networks (MLNs) (Richardson and Domingos, 2006) are among the most well-known methods for symbolic reasoning in relational data, which take advantage of the relational logic and probabilistic graphical models. They use the logic rules to define the potential function of a Markov random field (MRF) and thus soundly handle the uncertainty for the reasoning on various real-world tasks (Zhang et al., 2014; Poon and Domingos, 2007; Singla and Domingos, 2006b; Qu and Tang, 2019).

Typical methods perform inference in MLNs at the level of propositional logic via standard probabilistic inference methods, such as Gibbs sampling (MCMC) (Gilks et al., 1995; Richardson and Domingos, 2006), slice sampling (MC-SAT) (Poon and Domingos, 2006), belief propagation (Yedidia et al., 2000). However, the propositional probabilistic graph is extremely complicated as it typically generates all ground formulae as factors and the number is exponential in the arity of formulae. The methods need to manipulate these factors during the inference stage, leading to an exponential complexity. Besides, several lifted algorithms that treat the whole sets of indistinguishable objects identically were proposed, such as first-order variable elimination (de Salvo Braz et al., 2005), lifted BP (Singla and Domingos, 2008), lifted MCMC (Niepert, 2012). However, these lifted methods typically become infeasible when the symmetric structure breaks down, e.g., unique evidence is integrated for each variable. ExpressGNN (Zhang et al., 2020) uses a particular graph neural network as the posterior model in the variational EM training to amortize the problem of joint inference. However, its training complexity is still exponential in the length and arity of formulae. Overall, algorithm design for efficient MLN inference remains a challenge.

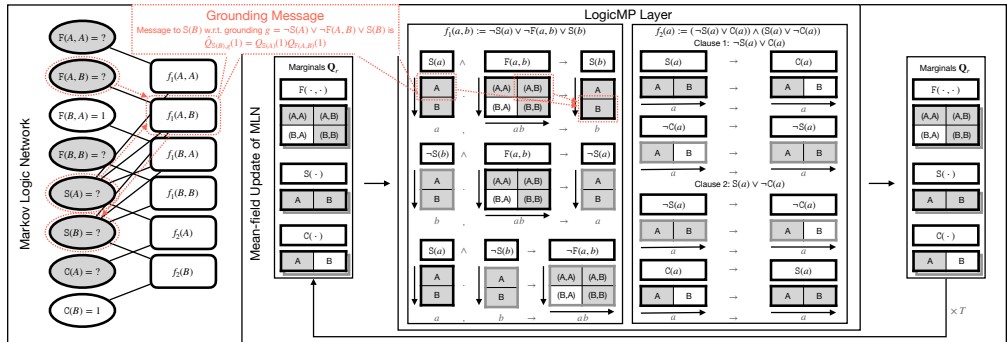

Figure 1: Left: A Markov logic network (MLN) with two entities $\{A, B\}$, predicates F (Friend) and S (Smoke) and C (Cancer), and formulae $f_1(a, b) := \neg S(a) \vee \neg F(a, b) \vee S(b)$, $f_2(a) := (\neg S(a) \vee C(a)) \wedge (S(a) \vee \neg C(a))$. The circles are the ground atoms where the shaded ones are latent and the blocks like $f_1(A, B)$ denote the ground formulae. Right: We expand the MLN inference into several mean-field iterations and each iteration is implemented by a LogicMP layer. The calculation of the message w.r.t. a single grounding is illustrated in the "grounding message" block with dashed lines: the message from $S(A)$ and $F(A, B)$ w.r.t. $f_1(A, B)$ can be simplified as $Q_{S(A)}(1)Q_{F(A,B)}(1)$. We then formulate parallel aggregation on the right via the Einstein summation (Einsum). Specifically, the marginals of ground atoms are grouped by the predicates as the basic units of computation (the gray border denotes the marginals with $\neg$). In each iteration, LogicMP takes them as input and performs Einsum for each logic rule (shown in the middle block). Intuitively, Einsum counts the groundings that derive the hypothesis for each implication statement of the logic rules (cf. Sec. 4). The outputs of Einsum are then used to update the grouped marginals. Such procedure loops for $T$ steps until convergence. Note that $f_2$ is in conjunctive normal form and its two clauses can be treated separately. The Einsum is also applicable for the predicates with more than two arguments.

To mitigate the inference problem of MLNs, we adopt the mean-field (MF) algorithm (Wainwright and Jordan, 2008; Koller and Friedman, 2009) to reason over the relational data approximately. The MF algorithm updates the marginals of variables by calculating the expected potential over other related variables. Although the iteration of the MF algorithm can be formulated as modern neural networks for several special conditional random fields (CRFs) (Zheng et al., 2015; Vemulapalli et al., 2016), its standard implementation of MLN is still inefficient due to expectation calculation over high-order related variables and massive groundings, resulting in a computational complexity exponential in the length and the arity of formulae, respectively. We show that the MF iteration in MLN can be efficiently formulated as a special neural network LogicMP, which propagates the messages for the logic rules especially. In most practical cases, for the logic rules in conjunctive normal form (CNF), it can remove both exponents to effectively reduce the complexity to polynomial.

Fig. 1 gives an overview of our approach with an example. For a given MLN (left), the MF algorithm unfolds the MLN inference into several iterations of the forward computation (right). Specifically, LogicMP tackles the two essential problems of the standard update from two perspectives. First, we show that in the MF update the expectation calculation over the high-order related variables of the logic rules can be removed owing to the property of CNF logic rules, and only one remaining state needs to consider which forms an implication path from the related variables (as the premise) to the updated variable (as the hypothesis). The block with the dashed line across both sides in Fig. 1 illustrates how to compute a message w.r.t. a single ground formula. Second, based on this finding, we further show that the computation and aggregation of massive groundings can be efficiently implemented by the Einstein summation (Einsum). It can not only aggregate the grounding messages in parallel but also indicate a way for complexity reduction. The Einsum operation can be optimized in general when the arguments are contracted sequentially. For instance, the MF iteration of chain rules can be implemented in cubic polynomial complexity regardless of the exponential amount of groundings in the arity of formulae. The experimental results on the benchmark datasets prove the effectiveness and efficiency of LogicMP, boosting the average AUC-PR by 11+% absolutely against many competitive methods with much faster training speed (about 10×) on the UW-CSE (Richardson and Domingos, 2006) and Cora (Singla and Domingos, 2005) datasets. Our reproducible source code is attached in the supplementary file and will be released publicly.

In summary, our main contributions include: (1) We propose that the MF inference for MLNs can be practically conducted at polynomial complexity for the CNF formulae in most cases, regardless of exponential propositional groundings in the formula arity. (2) We unify the formation of logic message passing for CNF via Einsum to enable parallel computation, and design a neural network layer, LogicMP, for MF iterations. (3) We verify LogicMP on four benchmark datasets of symbolic reasoning with considerable improvements in accuracy and efficiency. (4) Our released code supports multi-class scenarios that may benefit a broad range of applications with logical knowledge.

## 2 RELATED WORK

**Markov logic networks.** The research community in statistical relation learning (SRL) (Koller et al., 2007) combined the logic rules with the probabilistic models, giving rise to several methods (Neville and Jensen, 2007; Kersting and Raedt, 2008). MLNs (Richardson and Domingos, 2006) are among the most well-known methods proposed for SRL and have achieved remarkable results in various reasoning tasks (Poon and Domingos, 2007; Singla and Domingos, 2006b; Qu and Tang, 2019). They use logic rules to model the distribution of relational data as an MRF to absorb the noise. Despite the effort in improving the efficiency (de Salvo Braz et al., 2005; Singla and Domingos, 2006a; Poon and Domingos, 2006; Khot et al., 2011; Bach et al., 2017; Srinivasan et al., 2019; Jha et al., 2010; Singla and Domingos, 2008; Zhang et al., 2020; Venugopal et al., 2015; Sarkhel et al., 2016), the MLNs still struggle in efficient inference. Our method formulates the approximate MLN inference into the MF iterations so that the logical structured prediction can be directly achieved by multiple stacks of LogicMP layers.

**Mean-field algorithm.** The MF algorithm (Wainwright and Jordan, 2008; Koller and Friedman, 2009) is an approximate inference algorithm typically for the graphical models whose exact inference is intractable. It can be efficiently implemented for a fully connected pairwise CRF (Zheng et al., 2015; Krähenbühl and Koltun, 2013), Gaussian CRF (Vemulapalli et al., 2016) and linear-chain CRF (Wang et al., 2020). However, the MF algorithm still requires calculating the expected potential over the remaining related variables, which limits its application in MLNs where the graph structure is complicated due to the massive high-order connections constructed by the logic rules. The idea of unfolding the inference procedure goes beyond the MF algorithm such as Transformer for Hopfield network (Ramsauer et al., 2021) and ReduNet (Chan et al., 2021), which also motivate us to transform the process of symbolic reasoning into multiple steps of forward computations.

## 3 MARKOV LOGIC NETWORKS

An MLN is built upon the knowledge base $\{C, R, O\}$, consisting of three components, i.e., a set $C$ of constants, a set $R$ of predicates, and a set $O$ of observed facts. Each predicate $r$ is an indicator function $r(\cdot) : C \times ... \times C \mapsto \{0, 1\}$ to indicate whether the relation exists among the given constants.

With particular constants assigned to the predicate, we obtain the ground atom that is associated with a binary variable in probabilistic modeling. The MLN is defined over all such variables and a set of logic formulae $F$. Each formula $f$ is in the form of $f(\mathcal{A}) : C \times ... \times C \mapsto \{0, 1\}$ where $\mathcal{A}$ is its arguments. For instance, a formula in Fig. 1 with $\mathcal{A} = \{a, b\}$ is "$f(a, b) := \texttt{S}(a) \wedge \texttt{F}(a, b) \rightarrow \texttt{S}(b)$", which is equivalent to the disjunctive form "$\neg\texttt{S}(a) \vee \neg\texttt{F}(a, b) \vee \texttt{S}(b)$" by De Morgan's law.

For each formula $f$, we can obtain a set of groundings $G_f$ by assigning $\mathcal{A}$ with various constants in $C$. With the specific assignments to the arguments, the formula becomes the ground formula, aka grounding. For a grounding $g$, we use $\mathbf{v}_g$ to denote the variables associated with the ground atoms in $g$. For instance, a grounding $f(A, B) = \neg\texttt{S}(A) \vee \neg\texttt{F}(A, B) \vee \texttt{S}(B)$ can be obtained by substituting the arguments $(a, b)$ with constants $(A, B)$, $\mathbf{v}_g = (v_{\texttt{S}(A)}, v_{\texttt{F}(A,B)}, v_{\texttt{S}(B)})$ collects three involved variables. The ground formula can be seen as a function of $\mathbf{v}_g$ parameterized by $f$.

With the knowledge base and formulae, MLN can be generalized as follows:

$$p(\mathbf{v}|O) \propto \exp(\sum_i \phi_u(v_i) + \sum_{f \in F} w_f \sum_{g \in G_f} \phi_f(\mathbf{v}_g)), \tag{1}$$

where $\mathbf{v}$ is the collection of latent variables, $\phi_u(\cdot)$ is the independent unary potential of each ground atom $i$ which can be parameterized by other models, $w_f \in \mathcal{R}$ denotes a weight for the formula $f$, $\phi_f(\cdot)$ is the potential function defined by $f$ which simply takes the value of ground formula.

### 3.1 MEAN-FIELD UPDATE FOR MLNs

This paper focuses on the inference problem of MLN with the fixed MLN structure. Note that we consider all unobserved facts as the latent variables to infer under the open-word assumption. Since $p(\mathbf{v}|O)$ is generally intractable, we use the MF algorithm for approximate posterior marginal inference. Specifically, the MF algorithm computes a variational distribution $Q(\mathbf{v})$ that best approaches $p(\mathbf{v}|O)$ where $Q(\mathbf{v}) = \prod_i Q_i(v_i)$ is a product of independent marginal distributions over each latent variables. The algorithm minimizes the KL divergence $D_{KL}(Q(\mathbf{v})||p(\mathbf{v}|O))$:

$$\sum_{i,v_i} Q_i(v_i) \log Q_i(v_i) - \sum_{i,v_i} \phi_u(v_i)Q_i(v_i) - \sum_{f \in F} w_f \sum_{g \in G_f} \sum_{\mathbf{v}_g} \phi_f(\mathbf{v}_g) \prod_{i \in g} Q_i(v_i) + \log Z \,.$$

Note that minimizing $D_{KL}(Q(\mathbf{v})||p(\mathbf{v}|O))$ is equivalent to maximizing the evidence lower bound of $\log p(O)$. By considering $D_{KL}(Q(\mathbf{v})||p(\mathbf{v}|O))$ as a function of $Q_i(v_i)$, Wainwright and Jordan (2008) shows that the optimal $Q_i$ can be derived in closed-form as follows and the MF inference performs this update on each marginal $Q_i$ until convergence (cf. Appendix A):

$$Q_i(v_i) = \frac{1}{Z_i} \exp(\phi_u(v_i) + \sum_{f \in F} w_f \sum_{g \in G_f(i)} \hat{Q}_{i,g}(v_i)) \,, \tag{2}$$

where $Z_i$ is the partition function, $G_f(i)$ is the groundings of formula $f$ that involve ground atom $i$,

$$\hat{Q}_{i,g}(v_i) = \sum_{\mathbf{v}_{g-i}} \phi_f(v_i, \mathbf{v}_{g-i}) \prod_{j \in g_{-i}} Q_j(v_j) \tag{3}$$

is the grounding message of a single grounding $g$, and $g_{-i}$ denotes the ground atoms in the grounding $g$ except $i$. For instance, $g_{-\mathtt{S}(B)} = (\mathtt{S}(A), \mathtt{F}(A, B))$ removes $\mathtt{S}(B)$ from the variables in $f(A, B)$.

### 3.2 TIME COMPLEXITY ANALYSIS

To analyze the time complexity of an iteration by Eq. 2, we denote $N$ as the number of constants in $C$, $M = \max_f |\mathcal{A}^f|$ as the maximum arity of formulae, and $L = \max_f |f|$ as the maximum length (number of atoms) of formulae. For a single formula $\neg\mathtt{S}(a) \vee \neg\mathtt{F}(a, b) \vee \mathtt{S}(b)$, $M = 2$ and $L = 3$.

**Expectation calculation of grounding message.** The grounding message $\hat{Q}_{i,g}(v_i)$ represents the influence to the variable $i$ generated by the variables $g_{-i}$ w.r.t. the grounding $g$. The computation of grounding message in Eq. 3 needs to multiply $\prod_{j \in g_{-i}} Q_j(v_j)$ (which is $\mathcal{O}(L)$) for all possible values of $\mathbf{v}_{g-i}$ (which is $\mathcal{O}(2^{L-1})$), resulting in a complexity of $\mathcal{O}(L2^{L-1})$.

**Aggregation of massive groundings.** Since the number of groundings $|G_f|$ is $\mathcal{O}(N^M)$ and a grounding generates grounding messages for all the involved latent variables, we have $\mathcal{O}(N^M L)$ grounding messages. As the complexity of computing a grounding message is $\mathcal{O}(L2^{L-1})$, the total time complexity of an MF iteration in Eq. 2 is $\mathcal{O}(N^M L^2 2^{L-1})$, which is exponential in $M$ and $L$.

## 4 EFFICIENT MEAN-FIELD ITERATION VIA LOGICMP

In the following, we show that the exponent $L$ in the complexity can be removed by considering the property of the formulae in conjunctive normal form (CNF), and the exponent $M$ can also be reduced in general by optimizing the message aggregation via Einstein summation. In particular, the overall complexity can be reduced to $\mathcal{O}(N^3 L^2)$ for arbitrary chain rules.

### 4.1 LESS COMPUTATION PER GROUNDING MESSAGE

We first show the complexity of computing grounding message $\hat{Q}_{i,g}(v_i)$ in Eq. 3 can be reduced from $\mathcal{O}(L2^{L-1})$ to $\mathcal{O}(L)$ by simplifying the message calculation for the clauses, and then generalize it to the CNF formulae. The clauses are the basic formulae that can be expressed as the disjunction of literals (cf. Appendix B for more details). For convenience, we explicitly write the clause as $f(\cdot; \mathbf{n}^f)$ where $n_i^f$ is the preceding negation of atom $i$ in the clause $f$ ($f$ will be removed when the context is clean). For instance, $\neg\mathtt{S}(a) \vee \neg\mathtt{F}(a, b) \vee \mathtt{S}(b)$ is a clause and its $\mathbf{n}$ is $(1, 1, 0)$ for $(\mathtt{S}(a), \mathtt{F}(a, b), \mathtt{S}(b))$.

The following lemma states that in calculating $\hat{Q}_{i,g}(v_i)$ of Eq. 3 with the clauses, some particular values of $\mathbf{v}_{g-i}$ in $\sum_{\mathbf{v}_{g-i}}$ can be neglected. Conceptually, a clause has several equivalent implication statements and a grounding message can be seen as an implication from the premise of $g_{-i}$ to the hypothesis of $i$. When the particular value $\mathbf{v}_{g-i}^*$ makes the premise false, it can be neglected.

**Lemma 4.1.** *(No message of clause for the false premise.) When each formula $f(\cdot; \mathbf{n})$ is a clause, for a particular state $\mathbf{v}_{g-i}^*$ of a grounding $g \in G_f(i)$ that $\exists j \in g_{-i}, v_j^* = \neg n_j$, the MF iteration of Eq. 2 is equivalent for $\hat{Q}_{i,g}(v_i) = \sum_{\mathbf{v}_{g-i} \neq \mathbf{v}_{g-i}^*} \phi_f(v_i, \mathbf{v}_{g-i}) \prod_{j \in g_{-i}} Q_j(v_j)$.*

See proof in the Appendix C. This lemma leads to the following theorem (cf. proof in Appendix D):

**Theorem 4.2.** *(Message of clause considers true premise only.) When each formula $f(\cdot; \mathbf{n})$ is a clause, the MF iteration of Eq. 2 is equivalent for $\hat{Q}_{i,g}(v_i) = \mathbf{1}_{v_i = \neg n_i} \prod_{j \in g_{-i}} Q_j(v_j = n_j))$.*

Consequently, the grounding message can be simplified, e.g., $\hat{Q}_{\mathtt{S}(B),g}(1) = Q_{\mathtt{S}(A)}(1)Q_{\mathtt{F}(A,B)}(1)$. Compared to Eq. 3, the exponential summation of $\mathbf{v}_{g-i}$ is removed and the complexity of grounding message is reduced from $\mathcal{O}(L2^{L-1})$ to $\mathcal{O}(L)$ for the clauses. The theorem has a simple but important meaning: only when the premise is true, does the logic rule matter for the hypothesis. In fact, $\phi_f$ can be more sophisticated as long as the potentials make difference for $v_i$ only for the true premise. The CNF formulae are the conjunction of clauses, such as $f_2$ in Fig. 1. We show that the simplification can be generalized to CNF formulae as well for the $\mathcal{O}(L)$ complexity as follows.

**Theorem 4.3.** *(Message of CNF = $\sum$ message of clause.) When each formula $f$ is the conjunction of several distinct clauses $f_k(\cdot; \mathbf{n})$, the MF iteration of Eq. 2 is equivalent for $\hat{Q}_{i,g}(v_i) = \sum_{f_k} \mathbf{1}_{v_i = \neg n_i} \prod_{j \in g_{-i}} Q_j(v_j = n_j)$.*

See Appendix E for proof. From the theorem, we can see that the messages of several clauses can be computed separately and the message of CNF is equivalent to the summation of messages of its clauses with the same weight. Therefore, in Fig. 1, we can compute the message of CNF formula $f_2$ as two separate messages of its clauses. In the following, we only consider the clause formulae because the generalization to CNF is straightforward. In addition, we generalize the theorem for the formulae with multi-class predicates to benefit other tasks with logical knowledge (cf. Appendix F).

## 4.2 PARALLEL AGGREGATION USING EINSTEIN SUMMATION

This subsection provides an efficient formulation for the parallel message aggregation, i.e., summation of $G_f$ in Eq. 2, to reduce the exponent $M$. Naturally, we can generate all the propositional groundings in $G_f$ to perform the aggregation. However, the possible groundings can be massive, i.e., $\mathcal{O}(N^M)$. When $M$ is large, the graph may be very dense and such aggregation is infeasible in both space and time. Therefore, we propose a method to compute the messages via the Einstein summation (Einsum) which can not only aggregate grounding messages in parallel but also reduce complexity significantly for most practical formulae.

The virtue lies in the summation of the product, i.e., $\sum_{g \in G_f(i)} \prod_{j \in g_{-i}} Q_j(v_j = n_j)$ by Theorem 4.2. The theorem shows that the grounding message to a variable $i$ w.r.t. a clause is the probability of the premise of $g_{-i}$ being true. Note that a clause corresponds to several different implication statements and an implication statement corresponds to a kind of premise format in Theorem 4.2. The groundings in $G_f(i)$ w.r.t. variable $i$ may belong to different implication statements and therefore have different kinds of premise formats. Therefore, we group the grounding messages by the implications they belong to. The middle block of Fig. 1 gives a detailed depiction of the mechanism, where 3 clauses generate 7 implication statements and each first-order implication statement is transformed into an Einsum operation for parallel message computation. Let us consider the implication statement $\mathtt{S}(a) \wedge \mathtt{F}(a,b) \rightarrow \mathtt{S}(b)$. When $b$ is assigned as $B$, the groundings of the implication statement is $\{\mathtt{S}(a) \wedge \mathtt{F}(a,B) \rightarrow \mathtt{S}(B)\}_a$. We can aggregate the grounding messages of these groundings for $\mathtt{S}(B)$ by $\sum_a Q_{\mathtt{S}(a)}(1)Q_{\mathtt{F}(a,B)}(1)$. Note that the computation is also the same for various assignments of $b$ and $\sum_a Q_{\mathtt{S}(a)}(1)Q_{\mathtt{F}(a,b)}(1)$ can be expressed via Einsum as $\mathtt{einsum}(\text{"}a, ab \rightarrow b\text{"}, \mathbf{Q}_{\mathtt{S}}(\mathbf{1}), \mathbf{Q}_{\mathtt{F}}(\mathbf{1}))$, where $\mathbf{Q}_r(\mathbf{v}_r)$ denote the collection of marginals of predicate $r$, i.e., $\mathbf{Q}_r(\mathbf{v}_r) = \{Q_{r(\mathcal{A}_r)}(v_{r(\mathcal{A}_r)})\}_{\mathcal{A}_r}$ where $\mathcal{A}_r$ is the argument of $r$. Einsum also allows the computation for other complex formulae whose predicates have many arguments.

Formally, we now explicitly use $[f, i]$ to denote the implication statement of clause $f$ with $i$-th atom being the hypothesis for the convenience of aggregating the grounding messages in the same premise format, and then give the update equation via Einsum.

**Proposition.** *For the grounding messages w.r.t. the implication statement $[f, i]$ of a first-order clause $f(\mathcal{A}^f; \mathbf{n}^f)$ to its $i$-th atom, their aggregation is equivalent to the Einstein expression:*

$$\check{\mathbf{Q}}_{r_i}^{[f,i]}(\mathbf{v}_{r_i}) = \mathbf{1}_{\mathbf{v}_{r_i}=\neg n_i} \texttt{einsum}(\text{``}..., \mathcal{A}_{r_{j\neq i}}^f, ... \to \mathcal{A}_{r_i}^f\text{''}, ..., \mathbf{Q}_{r_{j\neq i}}(n_{j\neq i}), ...), \quad (4)$$

*where $r_i$ is the $i$-th predicate, $\mathcal{A}_{r_i}^f$ is the arguments of $r_i$. The MF iteration of Eq. 2 is equivalent to:*

$$\mathbf{Q}_r(\mathbf{v}_r) = \frac{1}{\mathbf{Z}_r} \exp(\Phi_u(\mathbf{v}_r) + \sum_{[f,i],r=r_i} w_f \check{\mathbf{Q}}_{r_i}^{[f,i]}(\mathbf{v}_{r_i})), \quad (5)$$

*where $\Phi_u(\mathbf{v}_r)$ is the collection of unary potentials of predicate $r$.*

Einsum can dramatically reduce the time complexity in some difficult situations. Let us consider a complicated chain rule with 4 arguments, $\texttt{I}(a,b) \wedge \texttt{F}(b,c) \wedge \texttt{G}(c,d) \to \texttt{S}(a,d)$ where $\texttt{I}, \texttt{F}, \texttt{G}, \texttt{S}$ are the predicates. The number of grounding messages is $\mathcal{O}(N^4)$. By Einsum, we can reduce $\mathcal{O}(N^4)$ to $\mathcal{O}(N^3)$. Specifically, $\check{\mathbf{Q}}_{\texttt{S}}^{[f,4]}(\mathbf{1}) = \texttt{einsum}(\text{``}ab, bc, cd \to ad\text{''}, \mathbf{Q}_{\texttt{I}}(\mathbf{1}), \mathbf{Q}_{\texttt{F}}(\mathbf{1}), \mathbf{Q}_{\texttt{G}}(\mathbf{1}))$ in the matrix notation, which can be optimized into two matrix multiplications in $\mathcal{O}(N^3)$. First, Einsum computes $\mathbf{Q}_{\texttt{I}}(\mathbf{1})\mathbf{Q}_{\texttt{F}}(\mathbf{1})$ which is $\mathcal{O}(N^3)$ to integrate the paths through $b$. Then it performs multiplication with $\mathbf{Q}_{\texttt{G}}(\mathbf{1})$ which is also $\mathcal{O}(N^3)$ to sum over $c$. Actually, the complexity is $\mathcal{O}(N^3)$ for any longer chain rules. In the practice, we use $\texttt{opt\_einsum}$ (Smith and Gray, 2018) to automatically optimize Einsum for arbitrary formulae. Notice that such implementation gives a unified formation of logic reasoning which covers the chain rules (Yang et al., 2017; Sadeghian et al., 2019) and the compositional rules (Yang and Song, 2020).

The optimization of Einsum is achieved by a search algorithm and the final complexity depends on the format of the formula and the order of arguments in the contraction. We show several Einsum cases and their optimal simplifications in Appendix G. In most cases, the complexity after the optimization is independent of $M$ and more improvement can be achieved for a more complex formula.

### 4.3 COMPLETING THE ITERATION

We complete our MF update algorithm in Algorithm 1. The algorithm implements the MF update by $T$ LogicMP layers. It takes the set of grouped unary potentials $\{\Phi_u(\mathbf{v}_r)\}_r$, formulae $\{f(\mathcal{A}; \mathbf{n})\}_f$ and their rule weights $\{w_f\}_f$ as input and outputs the updated marginals $\{\mathbf{Q}_r(\mathbf{v}_r)\}_r$ for all the predicates. First, an initial distribution of each latent variable is computed using the unary potentials via a normalization operation, i.e., the softmax layer. Second, we enumerate all the formulae and their implication statements to perform Einsum via Eq. 4 ($|f|$ is the length of $f$). Third, we combine the unary potentials and the outputs of Einsum to obtain a new estimation of marginals.

---

**Algorithm 1** MF update for MLN via LogicMP.

**Input:** Grouped unary potential $\{\Phi_u(\mathbf{v}_r)\}_r$, a set of formulae $\{f(\mathcal{A}; \mathbf{n})\}_f$ and their rule weights $\{w_f\}_f$, the number of LogicMP layers $T$.

$\mathbf{Q}_r(\mathbf{v}_r) \leftarrow \frac{1}{\mathbf{Z}_r} \exp(\Phi_u(\mathbf{v}_r)))$ for all predicates $r$.

**for** $t \in \{1, ..., T\}$ **do**         ▷ Layers
    **for** $f \in F$ **do**         ▷ Formulae
        **for** $i \in \{1, ..., |f|\}$ **do**    ▷ Implications
             Obtain $\check{\mathbf{Q}}_{r_i}^{[f,i]}(\mathbf{v}_{r_i})$ by Eq. 4.    ▷ Einsum
        **end for**
    **end for**
    Update $\mathbf{Q}_r(\mathbf{v}_r)$ by Eq. 5 for all predicates $r$.
**end for**
**return** $\{\mathbf{Q}_r(\mathbf{v}_r)\}_r$.

---

## 5 EXPERIMENTS

### 5.1 EXPERIMENTAL SETTINGS

**Benchmark datasets.** We verify LogicMP on the Smoke dataset (Badreddine et al., 2022) and three symbolic reasoning benchmark datasets. The Kinship (Zhang et al., 2020) dataset asks the questions in a relationship graph among the people. The social network dataset UW-CSE (Richardson and Domingos, 2006) contains information about students and professors in the CSE department of UW.

The entity resolution dataset Cora (Singla and Domingos, 2005) [1] consists of a collection of citations between academic papers. All datasets contain the specific formulae and all the formulae are in CNF. The details of datasets and general experimental settings are given in Appendix H.

**Evaluation metrics.** Following previous works (Richardson and Domingos, 2006; Singla and Domingos, 2005), we use the area under the precision-recall curve (AUC-PR) to evaluate the model performance. To evaluate the efficiency, we use the wall-clock time in minutes.

**Compared algorithms.** Our model is compared with several strong methods in the MLN literature, including MCMC (Gilks et al., 1995; Richardson and Domingos, 2006), belief propagation (BP) (Yedidia et al., 2000), lifted belief propagation (Lifted BP) (Singla and Domingos, 2008), MC-SAT (Poon and Domingos, 2006), hinge-loss Markov random field (HL-MRF) (Bach et al., 2017; Srinivasan et al., 2019) and variational EM (ExpressGNN) (Zhang et al., 2020).

**Method details.** In general, LogicMP performs MLN inference and can incorporate logic rules in various learning settings. Ideally, it can serve as a logic CRF for arbitrary neural networks and be trained end-to-end via maximum likelihood estimation (MLE) as in CRFasRNN (Zheng et al., 2015). However, as the labeled data is rare in our datasets, MLE is prone to overfitting and we turn to semi-supervised learning where rules act as priors to infer the latent variables.

To leverage the power of representation learning, LogicMP integrates with a light-weight neural predictor via the posterior regularization (Ganchev et al., 2010; Hu et al., 2016; Guo et al., 2018) to distill the inference results from LogicMP into the neural predictor in an iterative way. Specifically, the neural predictor is built upon the constants which are represented by a list of embedding vectors. Each predicate is modeled by a unique bi-linear layer which takes the concatenated constant vectors as input. LogicMP takes the output of the neural predictor as $\phi_u$ and performs symbolic reasoning to obtain estimated marginals. They in turn become the prediction targets in training the neural predictor. The detail of the method is attached in Appendix I.

We fix the formula weights $w_f$ of 1 and set the number of LogicMP layers $T$ to 5 (cf. Sec. 5.3) for all the experiments. Each experiment is performed 5 times and the average score is reported. As in ExpressGNN, we sample batches of groundings for the stochastic training (cf. Appendix J). Thanks to our acceleration techniques, we can perform large-scale training efficiently with a sampling size of 1024, i.e., batch size = 1024. Note that LogicMP can perform full graph computation efficiently, but the practical results indicate that training with sampling is more stable for the learning of logical knowledge for the neural predictor. We leave full graph training to future work. Note that Express-GNN also uses a special graph neural network to learn the logical knowledge, which approximately corresponds to the neural predictor in our approach. Different from their claim, we show the use of complex GNN is unnecessary in the large-scale training (cf. the model capacity in Appendix K).

## 5.2 MAIN RESULTS

**Smoke & Kinship.** The Smoke dataset is an example dataset to validate the usefulness of the proposed method and Fig. 2 shows that the correct results can be predicted. Table 1 demonstrates the AUC-PR results on the Kinship dataset, which is synthetic and noise-free with an increasing number of entities from S1 to S5. The compared methods are performed under the open-world setup, where unobserved facts are seen as latent variables (cf. Append L). '-' denotes that the method fails to complete the training as it is either out of memory or exceeds 24 hours. MCMC only manages to obtain the result on the smallest split. HL-MRF can achieve perfect accuracy in S1-S4 but it is invalid in the largest split. Our method can obtain nearly perfect results in all the splits, consistently outperforming all other competitors. These results demonstrate that our method is effective in precise reasoning.

**UW-CSE & Cora.** Table 2 shows the results of the UW-CSE dataset which is collected in a real-world scenario with considerable noise. In UW-CSE, the closed-world setup is used for the compared methods except for ExpressGNN as they are invalid under the open-world setup. Due to the noise in the dataset and the scarcity of observed facts, the results are much lower than those in Kinship and the competitor methods can hardly achieve an average AUC-PR of 20%. Even though, our method further improves the performance by an absolute margin of 11% against HL-MRF. Table 2 also shows the results of the Cora dataset, which is larger than the UW-CSE dataset. We can see

---

[1] This is not the Cora dataset (Sen et al., 2008) typically for the graph node classification.

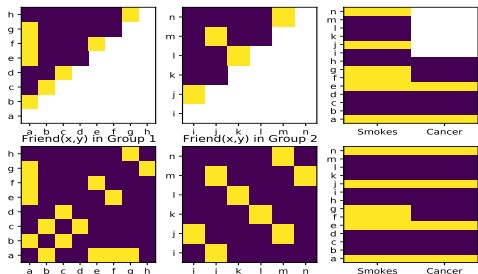

Figure 2: Facts (top) and predictions (bottom) on Smoke (🟨/🟪 for true/false facts).

Table 1: AUC-PR on Kinship. Best results are in bold. The bracket denotes the standard deviation.

| Method | Kinship | | | | |
|---|---|---|---|---|---|
| | S1 | S2 | S3 | S4 | S5 |
| MCMC | .53 | - | - | - | - |
| BP/Lifted BP | .53 | .58 | .55 | .55 | .56 |
| MC-SAT | .54 | .60 | .55 | .55 | - |
| HL-MRF | **1.0** | **1.0** | **1.0** | **1.0** | - |
| ExpressGNN | .97 | .97 | .99 | .99 | .99 |
| | (.01) | (.00) | (.01) | (.01) | (.00) |
| LogicMP | .99 | .98 | **1.0** | **1.0** | **1.0** |
| | (.00) | (.00) | (.00) | (.00) | (.00) |

Table 2: The AUC-PR on the UW-CSE and Cora datasets. The best results are in bold. The bracket denotes the standard deviation for which we rerun ExpressGNN 5 times. "A., G., L., S., T." are abbreviations for "AI, Graphics, Language, Systems, Theory".

| Method | UW-CSE | | | | | | Cora | | | | | |
|---|---|---|---|---|---|---|---|---|---|---|---|---|
| | A. | G. | L. | S. | T. | avg. | S1 | S2 | S3 | S4 | S5 | avg. |
| MCMC | .19 | .04 | .03 | .15 | .08 | .10 | .43 | .63 | .24 | .46 | .56 | .46 |
| BP/Lifted BP | .21 | .04 | .01 | .14 | .05 | .09 | .44 | .62 | .24 | .45 | .57 | .46 |
| MC-SAT | .13 | .04 | .03 | .11 | .08 | .08 | .43 | .63 | .24 | .46 | .57 | .47 |
| HL-MRF | .26 | .18 | .06 | .27 | .19 | .19 | .60 | .78 | .52 | .70 | .81 | .68 |
| ExpressGNN | .09 | .19 | .14 | .06 | .09 | .11 | .62 | .79 | .46 | .57 | .75 | .64 |
| | (.02) | (.02) | (.03) | (.02) | (.02) | (.02) | (.02) | (.01) | (.02) | (.03) | (.02) | (0.2) |
| LogicMP | **.25** | **.30** | **.42** | **.25** | **.28** | **.30** | **.80** | **.88** | **.72** | **.83** | **.89** | **.82** |
| | (.02) | (.04) | (.03) | (.02) | (.05) | (.03) | (.01) | (.01) | (.01) | (.01) | (.00) | (.01) |
| - batchsize=16 | .15 | .26 | .28 | .15 | .20 | .21 | .58 | .82 | .41 | .63 | .78 | .64 |
| | (.03) | (.04) | (.06) | (.02) | (.06) | (.04) | (.01) | (.01) | (.01) | (.01) | (.01) | (.01) |
| - nlayers=1 | .25 | .30 | .38 | .23 | .27 | .29 | .80 | .88 | .71 | .82 | .89 | .82 |
| | (.02) | (.04) | (.03) | (.02) | (.04) | (.03) | (.01) | (.01) | (.01) | (.01) | (.01) | (.01) |

that in the Cora dataset our method can also achieve better results against all the competitors and the improvements in the 5 splits are consistent. The performance in UW-CSE and Cora demonstrates the effectiveness of LogicMP in incorporating logic formulae in real-world tasks. We also visualize the test curves w.r.t. the training steps in Fig. 3 to demonstrate the robustness of our approach.

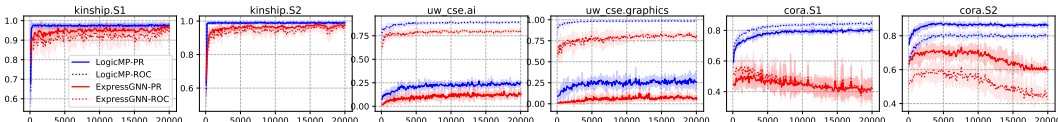

Figure 3: Test curves w.r.t. the training steps.

## 5.3 EFFECT OF MULTIPLE LOGICMP LAYERS

We show detailed ablation results with one LogicMP layer for the UW-CSE and Cora datasets in Table 2 and the ablation with various layers for all the splits of three datasets in Fig. 4. The experimental results show that the performance improves consistently when using multiple LogicMP layers. And the performance keeps stable when we further stack the layers. This is reasonable as the MF algorithm typically converges to a stable state within several steps. With more adequate interaction, the variables can gather more information from the logic formulae to make better reasoning, which also leads to better performance. Fortunately, LogicMP takes a few steps (typically within 5) to achieve good performance and we can empirically set $T$ to 5.

## 5.4 TRAINING EFFICIENCY

Table 2 shows the scalability capacity. From the ablation experiment against a small grounding batch size of 16, we can see that the performance improves considerably. The efficient implementa-

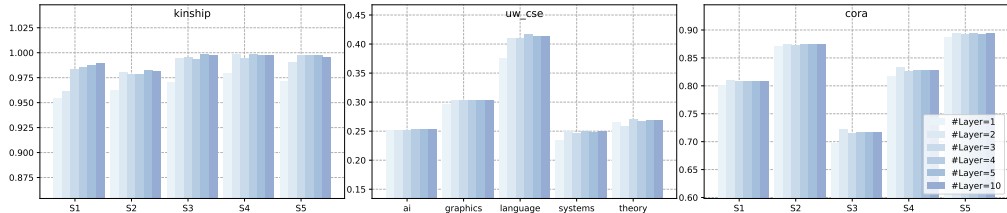

Figure 4: AUC-PR w.r.t. the number of LogicMP layers.

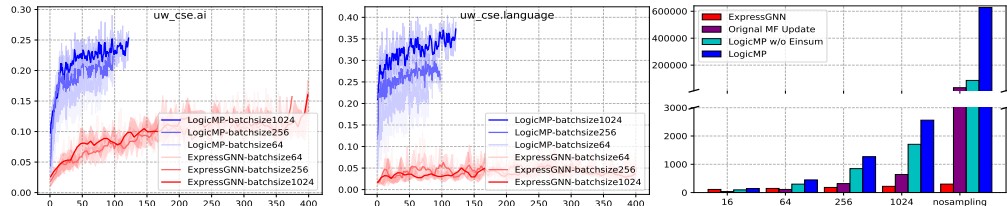

Figure 5: Left two: AUC-PR in minutes. Right: #Groundings/second w.r.t. batch size in Kinship.

tion of LogicMP is essential for large-scale training. We show the test AUC-PR curves of LogicMP and ExpressGNN w.r.t. training time on the left two of Fig. 5 on AI and Language cases of the UW-CSE dataset respectively. The learning method of ExpressGNN is computationally inefficient since it calculates ELBO with the expectation computation over posterior probabilities (which can be accelerated by the simplification in Sec. 4.1) and the groundings are computed separately. From the figure, we can see that the scores of ExpressGNN increase slowly and using more groundings in batches is ineffective, because the training speed consistently decreases for larger batch sizes without parallel implementation. In contrast, LogicMP can complete training in a rather short time with quickly increasing scores. When doubling the groundings, we can obtain better results at a faster speed. These results reveal the efficiency of LogicMP. For a detailed comparison of training efficiency, we ablate the grounding velocity of various implementations in the Kinship dataset on the right of Fig. 5. The original MF update denotes that the implementation without the technique in Sec. 4.1 or Einsum. For LogicMP w/o Einsum, we perform scattering and gathering to aggregate the propositional groundings (cf. Appendix M). We also used LogicMP to perform full graph computation which is demonstrated in the "nosampling" group. From the figure, we can see the MF update can outperform ExpressGNN remarkably and the two proposed techniques effectively accelerate the training speed, achieving $10\times$ training velocity than ExpressGNN (cf. Appendix N). When the sampling strategy is disabled, Einsum can calculate all the grounding messages in parallel and achieve superior efficiency.

## 6 CONCLUSION

We presented a novel neural network layer, LogicMP, to implement efficient MF update for MLNs. The efficiency improvement mainly comes from two non-trivial discoveries. We first showed that the grounding message of CNF is the summation of grounding messages of clauses that only need to consider the true premises, thus eliminating the exponent of the formulae length from computational complexity. Then, we showed that the aggregation of grounding messages can be transformed into Einstein summation for parallel computation, and Einsum itself can be optimized to reduce the exponent of the formulae arity in the complexity for most formulae in practice. Both contribute to the LogicMP layer that can perform MLN inference with polynomial complexity in general.

Note that LogicMP neglects the existential quantifier in the first-order logic. A plausible solution is to transform the formula into the Skolem norm form to eliminate the existential variables. It is also necessary to investigate the property of LogicMP both empirically and theoretically when serving as a logic CRF in the end-to-end training of some data-rich tasks. In the large knowledge bases, further reduction of cubic polynomial complexity is also worth exploring. We leave them to future work.

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

## A  MEAN-FIELD UPDATE EQUATION OF MLN

*Proof.* The conditional probability distribution of MLN in the inference problem is defined as:

$$p(\mathbf{v}|O) \propto \exp\left(\sum_i \phi_u(v_i) + \sum_{f \in F} w_f \sum_{g \in G_f} \phi_f(\mathbf{v}_g)\right). \tag{6}$$

$p(\mathbf{v}|O)$ is generally intractable since there is an exponential summation in the denominator. Therefore, we propose to use a proxy variational distribution $Q(\mathbf{v})$ to approximate the $p(\mathbf{v}|O)$ by minimizing the KL divergence $D_{KL}(Q(\mathbf{v})||p(\mathbf{v}|O))$. The proposed $Q(\mathbf{v})$ is an independent distribution over each variables, i.e., $Q(\mathbf{v}) = \prod_i Q_i(v_i)$ where $\sum_{v_i \in \{0,1\}} Q_i(v_i) = 1, Q_i(v_i) \geq 0$ is a proper probability.

Table 3: The used symbols and the corresponding denotations.

| Symbol | Definition |
|---|---|
| $f$ | the formula |
| $\|f\|$ | the number of atoms in the formula $f$ |
| $\mathcal{A}^f$ | the arguments of formula $f$ |
| $\|\mathcal{A}^f\|$ | the arity of formula $f$ |
| $g$ | the ground formula (grounding) |
| $O$ | the set of observed facts |
| $H$ | the set of unobserved facts |
| $v_i$ | the single variable associated with a ground atom $i$ |
| $G_f$ | the set of groundings of formula $f$ |
| $G_f(i)$ | the set of groundings of formula $f$ that contains $i$ |
| $\phi_u$ | the independent unary potential |
| $\Phi_u$ | the collection of unary potentials |
| $\phi_f$ | the potential of formula $f$ |
| $w_f$ | the weight of formula $f$ |
| $\mathbf{v}_g$ | the set of variables w.r.t. ground predicates in the grounding $g$ |
| $\mathbf{n}_f$ | the set of negations of ground predicates in the grounding $g$ |
| $Q_i$ | the marginal of variable $i$ |
| $\hat{Q}_{i,g}$ | the grounding message of grounding $g$ to the variable $i$ |
| $g_{-i}$ | the set of ground predicates in the grounding $g$ except $i$ |
| $\mathbf{v}_{g_{-i}}$ | the set of variables in the grounding $g$ except $i$ |
| $\mathbf{Q}_r$ | the collection of marginals of predicate $r$ |
| $[f,i]$ | the implication statement of formula $f$ to $i$-th atom |
| $\check{\mathbf{Q}}_{r_i}^{[f,i]}$ | the summation of grounding messages w.r.t. the implication $[f,i]$ |

Note that minimizing the KL divergence w.r.t. $Q(\mathbf{v})$ is equivalent to maximizing the evidence lower bound of $\log p(O)$:

$$
\begin{aligned}
D_{KL}(Q(\mathbf{v})\|p(\mathbf{v}|O)) &= \mathbb{E}_{Q(\mathbf{v})} \log \frac{Q(\mathbf{v})}{p(\mathbf{v}|O)} \\
&= \mathbb{E}_{Q(\mathbf{v})} \log Q(\mathbf{v}) - \mathbb{E}_{Q(\mathbf{v})} p(\mathbf{v}|O) \\
&= \mathbb{E}_{Q(\mathbf{v})} \log Q(\mathbf{v}) - \mathbb{E}_{Q(\mathbf{v})} p(\mathbf{v},O) + \log p(O) \\
&= -(\mathbb{E}_{Q(\mathbf{v})} p(\mathbf{v},O) - \mathbb{E}_{Q(\mathbf{v})} \log Q(\mathbf{v})) + \log p(O) \,,
\end{aligned}
\tag{7}
$$

which is the negative ELBO plus the log marginal probability of $O$ which is independent of $Q$.

Since $Q(\mathbf{v}) = \prod_i Q_i(v_i)$, we have:

$$
\mathbb{E}_{Q(\mathbf{v})} \log Q(\mathbf{v}) = \sum_i \sum_{v_i} Q_i(v_i) \log Q_i(v_i) \,.
\tag{8}
$$

and

$$
\mathbb{E}_{Q(\mathbf{v})} \log p(\mathbf{v}|O) = \sum_{i,v_i} \phi_u(v_i) Q_i(v_i) + \sum_{f \in F} w_f \sum_{g \in G_f} \sum_{\mathbf{v}_g} \phi_f(\mathbf{v}_g) \prod_{i \in g} Q_i(v_i) - \log Z \,,
\tag{9}
$$

where $Z$ is independent of $Q$.

We can therefore rewrite $D_{KL}(Q(\mathbf{v})\|p(\mathbf{v}|O))$ with these two equations as:

$$
\mathcal{L} = \sum_{i,v_i} Q_i(v_i) \log Q_i(v_i) - \sum_{i,v_i} \phi_u(v_i) Q_i(v_i) - \sum_{f \in F} w_f \sum_{g \in G_f} \sum_{\mathbf{v}_g} \phi_f(\mathbf{v}_g) \prod_{i \in g} Q_i(v_i) + \log Z \,.
\tag{10}
$$

Considering it as a function of $Q_i(v_i)$ and remove the irrelevant terms, we have:

$$\mathcal{L}_i = \sum_{v_i} Q_i(v_i) \log Q_i(v_i)$$
$$- \sum_{v_i} Q_i(v_i)[\phi_u(v_i) + \sum_{f \in F} w_f \sum_{g \in G_f(i)} \sum_{\mathbf{v}_{g-i}} \phi_f(v_i, \mathbf{v}_{g-i}) \prod_{j \in g-i} Q_j(v_i)], \quad (11)$$

where $g_{-i}$ is the ground variables except $i$ in the grounding $g$, $G_f(i)$ is the groundings of formula $f$ that involve ground atom $i$,

By the Lagrange multiplication theorem with the constraint that $\sum_{v_i} Q_i(v_i) = 1$, the problem becomes:

$$\arg \min_{Q_i(v_i)} \mathcal{L}'_i = \sum_{v_i} Q_i(v_i) \log Q_i(v_i)$$
$$- \sum_{v_i} Q_i(v_i)[\phi_u(v_i) + \sum_{f \in F} w_f \sum_{g \in G_f(i)} \sum_{\mathbf{v}_{g-i}} \phi_f(v_i, \mathbf{v}_{g-i}) \prod_{j \in g-i} Q_j(v_i)] \quad (12)$$
$$+ \lambda(\sum_{v_i} Q_i(v_i) - 1)$$

Take the derivative with respect to $Q_i(v_i)$:

$$\frac{d\mathcal{L}'_i}{dQ_i(v_i)} = 1 + \log Q_i(v_i) - [\phi_u(v_i) + \sum_{f \in F} w_f \sum_{g \in G_f(i)} \sum_{\mathbf{v}_{g-i}} \phi_f(v_i, \mathbf{v}_{g-i}) \prod_{j \in g-i} Q_j(v_i)] + \lambda. \quad (13)$$

Let the gradient be equal to 0, we then have:

$$Q_i(v_i) = \exp(\phi_u(v_i) + \sum_{f \in F} w_f \sum_{g \in G_f(i)} \sum_{\mathbf{v}_{g-i}} \phi_f(v_i, \mathbf{v}_{g-i}) \prod_{j \in g-i} Q_j(v_i) - 1 - \lambda), \quad (14)$$

Take $\lambda$ out from the equation, we have:

$$Q_i(v_i) = \frac{1}{Z_i} \exp(\phi_u(v_i) + \sum_{f \in F} w_f \sum_{g \in G_f(i)} \sum_{\mathbf{v}_{g-i}} \phi_f(v_i, \mathbf{v}_{g-i}) \prod_{j \in g-i} Q_j(v_j)), \quad (15)$$

where $Z_i$ is the partition function.

For clarity of presentation, we define the message of a single grounding (grounding message) as:

$$Q_i(v_i) = \frac{1}{Z_i} \exp(\phi_u(v_i) + \sum_{f \in F} w_f \sum_{g \in G_f(i)} \hat{Q}_{i,g}(v_i)),$$
$$\hat{Q}_{i,g}(v_i) = \sum_{\mathbf{v}_{g-i}} \phi_f(v_i, \mathbf{v}_{g-i}) \prod_{j \in g-i} Q_j(v_j). \quad (16)$$

Then we have the conclusion. □

## B  TERMINOLOGY

**Clause.** In logic, a clause is a formula formed from a finite disjunction of literals (atomic formulae or their negations). A clause is true whenever at least one of the literals that form it is true. Clauses are usually written as $(l_1 \vee l_2...)$, where the symbols $l_i$ are literals.

**Conjunctive Normal Form (CNF).** A formula is in conjunctive normal form if the formula is the conjunction of several clauses. The form is usually in $(l_{1,1} \vee l_{1,2}...) \wedge (l_{2,1} \vee l_{2,2}...) \wedge ...$, where the symbols $l_{i,j}$ are literals..

**First-order Logic.** First-order logic uses quantified variables over a set of constants and allows the use of sentences that contain variables, so that rather than propositions such as "Tom is a father, hence Tom is a man", one can have expressions in the form "for all person $x$, if $x$ is a father then $x$ is a man", where "for all" is a quantifier, while $x$ is a variable. In general, the clause and CNF are used for propositional logic. In this work, they denote the first-order clause and first-order CNF with universal quantifiers respectively.

**Ground Expression.** In mathematical logic, a ground term of a formal system is a term that does not contain any variables. Similarly, a ground formula is a formula that does not contain any variables. In this paper, we consider universal quantifiers, and a grounding of a formula denotes an assignment of constants to the arguments in the formula with universal quantifiers.

## C  PROOF OF LEMMA: NO MESSAGE OF CLAUSE FOR FALSE PREMISE

*Proof.* Let us consider a grounding $g^*$ in $G_f(i)$ and a grounding message from $g^*_{-i}$ to $i$ and there is a particular state $\mathbf{v}^*_{g^*_{-i}}$ that $\exists j \in g^*_{-i}, v^*_j = \neg n_j$:

$$
\begin{aligned}
Q_i(v_i) &= \frac{\exp(E_i(v_i))}{\sum_{v_i} \exp(E_i(v_i))}\,, \\
E_i(v_i) &= \phi_f(v_i, \mathbf{v}^*_{g^*_{-i}}; \mathbf{n}) \prod_{j \in g^*_{-i}} Q_j(v^*_j) + \Delta_i(v_i)\,, \\
\Delta_i(v_i) &= \phi_u(v_i) + \sum_{\mathbf{v}_{g^*_{-i}} \neq \mathbf{v}^*_{g^*_{-i}}} \phi_f(v_i, \mathbf{v}_{g^*_{-i}}; \mathbf{n}) \prod_{j \in g^*_{-i}} Q_j(v_j) \\
&\quad + \sum_f w_f \sum_{g \in G_f(i) \backslash g^*} \sum_{\mathbf{v}_{g_{-i}}} \phi_f(v_i, \mathbf{v}_{g_{-i}}; \mathbf{n}^f) \prod_{j \in g_{-i}} Q_j(v_j))\,.
\end{aligned}
\tag{17}
$$

Since $\exists j \in g^*_{-i}, v^*_j = \neg n^f_j$, the clause will always be true regardless of $v_i$, i.e., $\forall v_i \in \{0, 1\}, \phi_f(v_i, \mathbf{v}^*_{g^*_{-i}}) = 1$. Therefore, $\phi_f(v_i, \mathbf{v}^*_{g^*_{-i}}; \mathbf{n}^f) \prod_{j \in g^*_{-i}} Q_j(v^*_j)$ is independent of $v_i$:

$$
E_i(v_i) = C + \Delta_i(v_i)\,, C = w_f \prod_{j \in g^*_{-i}} Q_j(v^*_j)\,.
\tag{18}
$$

The two potentials of $C$ can be eliminated in the normalization step. We can apply the same logic to all the grounding messages and obtain the conclusion: $Q_i(v_i) = \frac{1}{Z_i} \exp(\phi_u(v_i) + \sum_f w_f \sum_{g \in G_f(i)} \hat{Q}_{i,g}(v_i))$, where $\hat{Q}_{i,g}(v_i) = \sum_{\mathbf{v}_{g_{-i}} \neq \mathbf{v}^*_{g_{-i}}} \phi_f(v_i, \mathbf{v}_{g_{-i}}) \prod_{j \in g_{-i}} Q_j(v_j)$.  □

## D  PROOF OF THEOREM: MESSAGE OF CLAUSE CONSIDERS TRUE PREMISE ONLY

*Proof.* By the lemma 4.1, only one remaining state needs to be considered, i.e., $v_j = n_j, \forall j \in g_{-i}$. And the potential $\phi_f(\cdot)$ is 1 iff $v_i = \neg n_i$, otherwise $\hat{Q}_{i,g}(v_i) \leftarrow 0$. Then we derive the conclusion: $Q_i(v_i) = \frac{1}{Z_i} \exp(\phi_u(v_i) + \sum_f w_f \sum_{g \in G_f(i)} \hat{Q}_{i,g}(v_i))$, where $\hat{Q}_{i,g}(v_i) = \mathbf{1}_{v_i = \neg n^f_i} \prod_{j \in g_{-i}} Q_j(v_j = n^f_j)$.  □

# E    PROOF OF THEOREM: MESSAGE OF CNF = $\sum$ MESSAGE OF CLAUSE

*Proof.* For convenience, let us consider a grounding $g^*$ in $G_f(i)$ where $f$ in CNF is the conjunction of several distinct clauses $f_k(\cdot; \mathbf{n}^{f_k})$:

$$
\begin{aligned}
Q_i(v_i) &= \frac{\exp(E_i(v_i))}{\sum_{v_i} \exp(E_i(v_i))}\,, \\
E_i(v_i) &= \sum_{\mathbf{v}_{g^*_{-i}}} \phi_f(v_i, \mathbf{v}_{g^*_{-i}}) \prod_{j \in g^*_{-i}} Q_j(v_j) + \Delta_i(v_i)\,, \\
\Delta_i(v_i) &= \phi_u(v_i) + \sum_f w_f \sum_{g \in G_f(i) \backslash g^*} \sum_{\mathbf{v}_{g_{-i}}} \phi_f(v_i, \mathbf{v}_{g_{-i}}) \prod_{j \in g^*_{-i}} Q_j(v_j))\,.
\end{aligned}
\tag{19}
$$

Let $\mathbf{v}_{g^*_{-i}}^k$ be $\{n_j^{f_k}\}_{j \in g^*_{-i}}$, i.e., the corresponding true premises of clauses. We have:

$$
E_i(v_i) = \sum_k \phi_f(v_i, \mathbf{v}_{g^*_{-i}}^k) \prod_{j \in \mathbf{v}_{g^*_{-i}}^k} Q_j(v_j^k) + \sum_{\mathbf{v}_{g^*_{-i}} \notin \{\mathbf{v}_{g^*_{-i}}^k\}} \phi_f(v_i, \mathbf{v}_{g^*_{-i}}) \prod_{j \in g^*_{-i}} Q_j(v_j) + \Delta_i(v_i)\,, \tag{20}
$$

where the second term can be directly eliminated as in the proof of Lemma 4.1. We consider two cases that $\mathbf{v}_{g^*_{-i}}^k$ is unique or not for various $k$.

- Case 1 $\mathbf{v}_{g^*_{-i}}^k$ is unique: Since $\mathbf{v}_{g^*_{-i}}^k$ is unique, then $\phi_f(v_i, \mathbf{v}_{g^*_{-i}}^k) = \mathbf{1}_{v_i = \neg n_i^{f_k}}$ and we can get the message by the same logic in Theorem 4.2, i.e., $\mathbf{1}_{v_i = \neg n_i^{f_k}} \prod_{j \in g^*_{-i}} Q_j(v_j = n_j^{f_k})$.

- Case 2 $\mathbf{v}_{g^*_{-i}}^k$ is not unique: Let $\mathbf{v}_{g^*_{-i}}^{k_1}$ and $\mathbf{v}_{g^*_{-i}}^{k_2}$ be the same where $k_1$ and $k_2$ are two clauses in $f$. Since the clauses are unique, $n_i^{f_{k_1}}$ must be different with $n_i^{f_{k_2}}$. The two potentials will eliminated, which is equivalent to the summation of distinct messages, i.e., $\sum_{k_1, k_2} \mathbf{1}_{v_i = \neg n_i^{f_k}} \prod_{j \in g^*_{-i}} Q_j(v_j = n_j^{f_k})$.

We can apply this logic for every possible groundings and obtain: $Q_i(v_i) = \frac{1}{Z_i} \exp(\phi_u(v_i) + \sum_f w_f \sum_{g \in G_f(i)} \hat{Q}_{i,g}(v_i)$, where $\hat{Q}_{i,g}(v_i) = \sum_k \mathbf{1}_{v_i = \neg n_i^{f_k}} \prod_{j \in g_{-i}} Q_j(v_j = n_j^{f_k})$, i.e., $\sum_{f_k} \mathbf{1}_{v_i = \neg n_i} \prod_{j \in g_{-i}} Q_j(v_j = n_j)$. $\square$

This theorem directly leads to the following corollary:

**Corollary E.1.** *For the MLN, the mean-field update w.r.t. a CNF formula is equivalent to the mean-field update w.r.t. multiple clause formulae with the same rule weight.*

# F    EXTENSION OF MULTI-CLASS PREDICATES

A typical Markov logic network is defined over binary variables where the corresponding fact can be either true or false. This is unusual in the modeling of common tasks, where the exclusive categories form a single multi-class classification. For instance, a typical MLN will use several distinct binary predicates to describe the category of a paper. However, the categories are typically exclusive, and combining them can ease model learning. Therefore, we extend the LogicMP model to the multi-class predicates.

Formally, let the predicates be multi-class classifications $r(\cdot) : C \times ... \times C \to \{0, 1, ...\}$ with $\geq 2$ categories, which is different with standard MLN. The atom in the formula is then equipped with another configuration $\mathcal{Z}$ for the valid value of predicates. For instance, a multi-class formula about "RL paper cites RL paper" can be expressed as "$P(x) \in \{1\} \vee C(x, y) \in \{1\} \to P(y) \in \{1\}$" ($P(x) = 1$ means $x$ is a RL paper, $C(x, y) = 1$ means $x$ cites $y$). Sometimes the predicates in the formula appears more than one time, e.g., $P(x) \in \{2\} \vee P(x) \in \{1\}$.... We should aggregate them into a single literal $P(x) \in \mathcal{Z}, \mathcal{Z} = \{1, 2\}$. A clause with multi-class predicates is then formulated

as $... \vee (v_i \in \mathcal{Z}_i) \vee ...$ where $v_i$ is the variable associated with the atom $i$ in the clause and $\mathcal{Z}_i$ denotes the possible values the predicate can take. In such notation, we rewrite the clauses with multi-class predicates as $f(\cdot; \mathcal{Z}^f)$ where $\mathcal{Z}^f = \{\mathcal{Z}_i\}_i$. We show that the message of the multi-class clause can be derived by the following theorem.

**Theorem F.1.** *When each formula with multi-class predicates $f(\cdot; \mathcal{Z}^f)$ is a clause, the MF iteration of Eq. 2 is equivalent for $\hat{Q}_i(v_i) = \mathbf{1}_{v_i \in \mathcal{Z}_i} \prod_{j \in g_{-i}} (1 - \sum_{v_j \in \mathcal{Z}_j} Q_j(v_j))$.*

As the derivation is similar to that of binary predicates, we omit the detailed proof here. By setting $\mathcal{Z} = \{\neg n_i\}$ in the binary case, we can see that the message with multi-class predicates becomes the one with binary predicates. Similarly, when the formula is the CNF, the message can be calculated by aggregating the messages of clauses as in Theorem 4.3.

## G  THE EINSTEIN SUMMATION

The Einstein summation [2] is the notation for the summation of the product of elements in a list of high-dimensional tensors. We found the aggregation of grounding messages w.r.t. an implication statement can be exactly represented by an Einstein summation expression. And the Einstein summation can be efficiently implemented in parallel via NumPy and nowadays deep learning frameworks, e.g., PyTorch and TensorFlow. The corresponding function is called `einsum` [3] which can be effectively optimized via a library `opt_einsum` [4]. We list several cases of message aggregation in Einstein summation format and their optimal simplifications via dynamic argument contraction.

Formula: $\text{R}_1(h,k) \wedge \text{R}_2(k,j) \wedge \text{R}_3(j,i) \rightarrow \text{R}_0(i)$

- Original: $\mathbf{K} \leftarrow \texttt{einsum}(\text{``}hk, kj, ji \rightarrow i\text{''}, \mathbf{Q}_{\text{R}_1}(\mathbf{1}), \mathbf{Q}_{\text{R}_2}(\mathbf{1}), \mathbf{Q}_{\text{R}_3}(\mathbf{1}))$
- Optimized:
    - $\mathbf{K} \leftarrow \texttt{einsum}(\text{``}kj, ji \rightarrow ki\text{''}, \mathbf{Q}_{\text{R}_2}(\mathbf{1}), \mathbf{Q}_{\text{R}_3}(\mathbf{1}))$
    - $\mathbf{K} \leftarrow \texttt{einsum}(\text{``}hk, ki \rightarrow i\text{''}, \mathbf{Q}_{\text{R}_1}(\mathbf{1}), \mathbf{K})$

Formula: $\text{R}_1(h,k) \wedge \text{R}_2(k,j) \wedge \text{R}_3(j,i) \wedge \text{R}_4(h) \rightarrow \text{R}_0(i)$

- Original: $\mathbf{K} \leftarrow \texttt{einsum}(\text{``}hk, kj, ji, h \rightarrow i\text{''}, \mathbf{Q}_{\text{R}_1}(\mathbf{1}), \mathbf{Q}_{\text{R}_2}(\mathbf{1}), \mathbf{Q}_{\text{R}_3}(\mathbf{1}), \mathbf{Q}_{\text{R}_4}(\mathbf{1}))$
- Optimized:
    - $\mathbf{K} \leftarrow \texttt{einsum}(\text{``}kj, ji \rightarrow ki\text{''}, \mathbf{Q}_{\text{R}_2}(\mathbf{1}), \mathbf{Q}_{\text{R}_3}(\mathbf{1}))$
    - $\mathbf{K} \leftarrow \texttt{einsum}(\text{``}hk, h, ki \rightarrow i\text{''}, \mathbf{Q}_{\text{R}_1}(\mathbf{1}), \mathbf{Q}_{\text{R}_4}(\mathbf{1}), \mathbf{K})$

Formula: $\text{R}_1(p,i) \wedge \text{R}_1(q,j) \wedge \text{R}_2(i,j,k,l) \wedge \text{R}_1(r,k) \wedge \text{R}_1(s,l) \rightarrow \text{R}_0(p,q,r,s)$

- Original: $\mathbf{K} \leftarrow \texttt{einsum}(\text{``}pi, qj, ijkl, rk, sl \rightarrow pqrs\text{''}, \mathbf{Q}_{\text{R}_1}(\mathbf{1}), \mathbf{Q}_{\text{R}_1}(\mathbf{1}), \mathbf{Q}_{\text{R}_2}(\mathbf{1}), \mathbf{Q}_{\text{R}_1}(\mathbf{1}), \mathbf{Q}_{\text{G}}(\mathbf{1}))$
- Optimized:
    - $\mathbf{K} \leftarrow \texttt{einsum}(\text{``}pi, ijkl \rightarrow pjkl\text{''}, \mathbf{Q}_{\text{R}_1}(\mathbf{1}), \mathbf{Q}_{\text{R}_2}(\mathbf{1}))$
    - $\mathbf{K} \leftarrow \texttt{einsum}(\text{``}qj, pjkl \rightarrow pqkl\text{''}, \mathbf{Q}_{\text{R}_1}(\mathbf{1}), \mathbf{K})$
    - $\mathbf{K} \leftarrow \texttt{einsum}(\text{``}rk, pqkl \rightarrow pqrl\text{''}, \mathbf{Q}_{\text{R}_1}(\mathbf{1}), \mathbf{K})$
    - $\mathbf{K} \leftarrow \texttt{einsum}(\text{``}sl, pqrl \rightarrow pqrs\text{''}, \mathbf{Q}_{\text{R}_1}(\mathbf{1}), \mathbf{K})$

We also show several cases that cannot be optimized:

Formula: $\text{R}_1(a) \wedge \text{R}_2(a,b) \rightarrow \text{R}_1(b)$

- $\mathbf{K} \leftarrow \texttt{einsum}(\text{``}a, ab \rightarrow b\text{''}, \mathbf{Q}_{\text{R}_1}(\mathbf{1}), \mathbf{Q}_{\text{R}_2}(\mathbf{1}))$

Formula: $\text{R}_1(a,b,c,d) \wedge \text{R}_2(b,c) \wedge \text{R}_3(c,d) \wedge \text{R}_4(a,d) \rightarrow \text{R}_0(a,c)$

- $\mathbf{K} \leftarrow \texttt{einsum}(\text{``}abcd, bc, cd, ad \rightarrow ac\text{''}, \mathbf{Q}_{\text{R}_1}(\mathbf{1}), \mathbf{Q}_{\text{R}_2}(\mathbf{1}), \mathbf{Q}_{\text{R}_3}(\mathbf{1}), \mathbf{Q}_{\text{R}_4}(\mathbf{1}))$

---

[2] https://en.wikipedia.org/wiki/Einstein_notation

[3] https://pytorch.org/docs/stable/generated/torch.einsum.html

[4] https://optimized-einsum.readthedocs.io/en/stable/

Formula: $\mathtt{R_1}(a,b,c) \wedge \mathtt{R_2}(b,c,d) \wedge \mathtt{R_3}(c,b) \wedge \mathtt{R_4}(a,d) \rightarrow \mathtt{R_0}(a,c)$

- $\mathbf{K} \leftarrow \mathtt{einsum}(\text{``}abc, bcd, cb, ad \rightarrow ac\text{''}, \mathbf{Q}_{\mathtt{R_1}}(\mathbf{1}), \mathbf{Q}_{\mathtt{R_2}}(\mathbf{1}), \mathbf{Q}_{\mathtt{R_3}}(\mathbf{1}), \mathbf{Q}_{\mathtt{R_4}}(\mathbf{1}))$

One may notice that the current implementations of the Einsum function are not available when the target matrix has external arguments that are not in the input matrices, i.e., "$a \rightarrow ab$". We tackle this by a post-processing function for the output of Einsum.

## H    MORE EXPERIMENTAL SETTINGS

**Prediction tasks.** There are several predicates in each dataset. In the Kinship dataset, the prediction task is to answer the gender of the person in the query, e.g., $\mathtt{male}(c)$, which can be inferred from the relationship between the persons. For instance, a person can be deduced as a male by the fact that he is the father of someone and the formula expressing a father is male. In the UW-CSE dataset, we need to infer $\mathtt{AdvisedBy}(a, b)$ when the facts about teachers and students are given. The dataset is split into five sets according to the home department of the entities. The Cora dataset contains the queries to de-duplicate entities, and one of the queries is $\mathtt{SameTitle}(a, b)$. The dataset is also split into five subsets according to the field of research.

**Statistics.** The details of the benchmark datasets are illustrated in Table 4.

Table 4: The details of the benchmark datasets.

| Dataset | #entity | #relation | #fact | #query | #ground predicate | #ground formula |
|---|---|---|---|---|---|---|
| Kinship/S1 | 62 | 15 | 187 | 38 | 50K | 550K |
| Kinship/S2 | 110 | 15 | 307 | 62 | 158K | 3M |
| Kinship/S3 | 160 | 15 | 482 | 102 | 333K | 9M |
| Kinship/S4 | 221 | 15 | 723 | 150 | 635K | 23M |
| Kinship/S5 | 266 | 15 | 885 | 183 | 920K | 39M |
| UW-CSE/AI | 300 | 22 | 731 | 4K | 95K | 73M |
| UW-CSE/Graphics | 195 | 22 | 449 | 4K | 70K | 64M |
| UW-CSE/Language | 82 | 22 | 182 | 1K | 15K | 9M |
| UW-CSE/Systems | 277 | 22 | 733 | 5K | 95K | 121M |
| UW-CSE/Theory | 174 | 22 | 465 | 2K | 51K | 54M |
| Cora/S1 | 670 | 10 | 11K | 2K | 175K | 621B |
| Cora/S2 | 602 | 10 | 9K | 2K | 156K | 431B |
| Cora/S3 | 607 | 10 | 18K | 3K | 156K | 438B |
| Cora/S4 | 600 | 10 | 12K | 2K | 160K | 435B |
| Cora/S5 | 600 | 10 | 11K | 2K | 140K | 339B |

**Formulae of the datasets.** We show several logic rules in the datasets in Table 5. The blocks each of which contains 5 rule examples correspond to the Smoke, Kinship, UW-CSE, and Cora datasets. The maximum length of Smoke and Kinship rules is 3, and 6 for the UW-CSE and Cora datasets. We can see from the table that all the logic formulae are CNF. Note that some formulae contain fixed constants such as "Post_Quals" and "Level_100" and we should not treat them as arguments.

**General settings.** The experiments were conducted on a basic machine with a 16GB P100 GPU and an Intel E5-2682 v4 CPU at 2.50GHz with 32GB RAM. The model is trained with the Adam optimizer with a learning rate of 5e-4.

## I    INCORPORATING LOGIC RULES VIA SEMI-SUPERVISED LEARNING

LogicMP can be used as a component to integrate logic rules in various learning settings. As the labeled data is much rare compared to the latent variables, the usage like CRFasRNN via end-to-end supervised learning is prone to overfitting for our tasks and hence we adopt a semi-supervised technique, i.e., posterior regularization (Ganchev et al., 2010; Hu et al., 2016; Guo et al., 2018), to incorporate the logical knowledge. It uses the logic rules as the prior constraints to guide the model via distillation. Fig. 6 gives an illustration of our approach. In the figure, LogicMP is stacked upon a neural predictor with parameters $\theta$ to take advantage of both worlds (the symbolic ability of LogicMP and the semantic ability of the neural predictor). The neural predictor is responsible

Table 5: Several logic rules in the datasets. # P denotes the number of predicates.

| First-order Logic Formula | # P |
|---|---|
| $\neg\texttt{smoke}(a) \vee \neg\texttt{friend}(a,b) \vee \texttt{smoke}(b)$ | 3 |
| $\neg\texttt{smoke}(a) \vee \texttt{cancer}(a)$ | 2 |
| $\texttt{smoke}(a) \vee \neg\texttt{cancer}(a)$ | 2 |
| $\neg\texttt{friend}(a,a)$ | 1 |
| $\neg\texttt{friend}(a,b) \vee \texttt{friend}(a,b)$ | 2 |
| $\neg\texttt{female}(x) \vee \neg\texttt{child}(y,x) \vee \texttt{mother}(x,y)$ | 3 |
| $\neg\texttt{male}(x) \vee \neg\texttt{child}(y,x) \vee \texttt{father}(x,y)$ | 3 |
| $\neg\texttt{female}(x) \vee \neg\texttt{child}(x,y) \vee \texttt{daughter}(x,y)$ | 3 |
| $\neg\texttt{male}(x) \vee \neg\texttt{child}(x,y) \vee \texttt{son}(x,y)$ | 3 |
| $\neg\texttt{male}(x) \vee \neg\texttt{female}(x)$ | 2 |
| $\neg\texttt{taughtBy}(c,p,q) \vee \neg\texttt{courseLevel}(c,\texttt{Level\_500}) \vee \neg\texttt{ta}(c,s,q) \vee \texttt{advisedBy}(s,p) \vee \texttt{tempAdvisedBy}(s,p)$ | 5 |
| $\neg\texttt{publication}(p,x) \vee \neg\texttt{publication}(p,y) \vee \neg\texttt{student}(x) \vee \texttt{student}(y) \vee \texttt{advisedBy}(x,y) \vee \texttt{tempAdvisedBy}(x,y)$ | 5 |
| $\neg\texttt{inPhase}(s,\texttt{Post\_Quals}) \vee \neg\texttt{taughtBy}(c,p,q) \vee \neg\texttt{ta}(c,s,q) \vee \texttt{courseLevel}(c,\texttt{Level\_100}) \vee \texttt{advisedBy}(s,p)$ | 5 |
| $\neg\texttt{student}(x) \vee \texttt{advisedBy}(x,y) \vee \texttt{tempAdvisedBy}(x,y)$ | 3 |
| $\neg\texttt{publication}(t,a) \vee \neg\texttt{publication}(t,b) \vee \texttt{samePerson}(a,b) \vee \texttt{advisedBy}(a,b) \vee \texttt{advisedBy}(b,a)$ | 5 |
| $\neg\texttt{Author}(bc1,a1) \vee \neg\texttt{Author}(bc2,a2) \vee \neg\texttt{HasWordAuthor}(a1,+w) \vee \neg\texttt{HasWordAuthor}(a2,+w) \vee \texttt{SameBib}(bc1,bc2)$ | 5 |
| $\neg\texttt{Author}(bc1,a1) \vee \neg\texttt{Author}(bc2,a2) \vee \texttt{HasWordAuthor}(a1,+w) \vee \neg\texttt{HasWordAuthor}(a2,+w) \vee \texttt{SameBib}(bc1,bc2)$ | 5 |
| $\neg\texttt{Title}(bc1,t1) \vee \neg\texttt{Title}(bc2,t2) \vee \neg\texttt{HasWordTitle}(t1,+w) \vee \neg\texttt{HasWordTitle}(t2,+w) \vee \texttt{SameBib}(bc1,bc2)$ | 5 |
| $\neg\texttt{Title}(bc1,t1) \vee \neg\texttt{Title}(bc2,t2) \vee \texttt{HasWordTitle}(t1,+w) \vee \neg\texttt{HasWordTitle}(t2,+w) \vee \texttt{SameBib}(bc1,bc2)$ | 5 |
| $\neg\texttt{Venue}(bc1,v1) \vee \neg\texttt{Venue}(bc2,v2) \vee \neg\texttt{HasWordVenue}(v1,+w) \vee \neg\texttt{HasWordVenue}(v2,+w) \vee SameBib(bc1,bc2)$ | 5 |

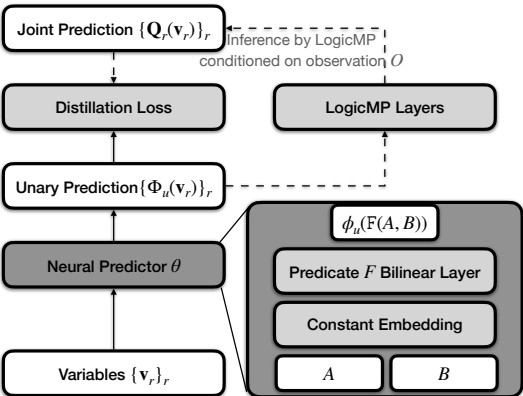

Figure 6: The illustration of incorporating the logic rules into the models via posterior regularization. In this learning paradigm, LogicMP plays the role of logical inference layer for the neural predictor with parameters $\theta$. Specifically, we are given a set of variables $\mathbf{v}$. The neural predictor takes the variables as input and output scores $\Phi_u$ as the unary potentials of variables. Then the unary potentials are fed into the LogicMP layers to perform MF inference to derive updated marginals $\mathbf{Q}$. They in turn become the targets of unary prediction through the distillation loss. The dotted line means no gradients in the back-propagation. The right bottom block gives the details of the neural predictor: for each variable of the ground atom, it converts the constants into a list of embedding vectors. The concatenation of the vectors is fed into a bilinear layer to obtain the unary potential.

for estimating the unary potentials of the variables independently. Then the LogicMP layers take these unary potentials to perform MF inference which derives a more accurate prediction with the constraints of logic rules. The outputs of LogicMP layers then become the targets of the neural predictor through a distillation loss for better estimation. In this way, logical knowledge can be distilled into the neural predictor. Note that LogicMP in this process only performs inference without any learning. Intuitively, the neural predictor is responsible for point estimation, which LogicMP gives the joint estimation via symbolic reasoning. LogicMP helps to adjust the distribution of the neural predictor since the output of LogicMP can not only match the original prediction but also fit the logic rules.

We show the derivation of such a learning paradigm from the posterior regularization as follows. Specifically, the posterior regularization method derives another target distribution $h(\mathbf{v})$ by minimizing $D_{KL}(h(\mathbf{v})||p_\theta(\mathbf{v}|O))$ with the prior constraints $\phi$ from the logic rules. Ganchev et al. (2010) shows that the optimal $h(\mathbf{v})$ can be obtained in the closed form: $h(\mathbf{v}) \sim p_\theta(\mathbf{v}|O) \exp(\lambda\phi(\mathbf{v}, O))$,

Table 6: The comparison of AUC-PR with different $p_g$. The bracket denotes the standard deviation. "A., G., L., S., T." are abbreviations for "AI, Graphics, Language, Systems, Theory".

| | UW-CSE | | | | | | Cora | | | | | |
|---|---|---|---|---|---|---|---|---|---|---|---|---|
| | A. | G. | L. | S. | T. | avg. | S1 | S2 | S3 | S4 | S5 | avg. |
| $p_g = 0.0$ | .08 | .13 | .44 | .06 | .13 | .17 | .44 | .62 | .25 | .46 | .57 | .47 |
| | (.01) | (.04) | (.12) | (.02) | (.13) | (.06) | (.02) | (.01) | (.01) | (.01) | (.02) | (.01) |
| $p_g = 0.9$ | .25 | .30 | .42 | .25 | .28 | .30 | .80 | .88 | .72 | .83 | .89 | .82 |
| | (.02) | (.04) | (.03) | (.02) | (.05) | (.03) | (.01) | (.01) | (.01) | (.01) | (.00) | (.01) |

where $\lambda$ is a hyper-parameter. When the constraint $\phi(\mathbf{v}, O) = \sum_{f \in F} w_f \sum_{g \in G_f} \phi_f(\mathbf{v}_g)$ (Markov logic) and $p_\theta(\mathbf{v}|O) \sim \exp(\sum_i \phi_u(v_i; \theta))$ (neural predictor), we have $h(\mathbf{v}) \sim \exp(\sum_i \phi_u(v_i; \theta) + \lambda \sum_{f \in F} w_f \sum_{g \in G_f} \phi_f(\mathbf{v}_g))$ which is equivalent to our first definition in Eq. 1. We want to distill $h(\mathbf{v})$ to $p_\theta(\mathbf{v}|O)$. However, since the target distribution is an MRF, direct distillation is difficult. Following the work (Wang et al., 2021), we calculate the marginal of target distribution for each variables $Q_i(v_i)$ via LogicMP and distill the knowledge by minimizing the distance between local marginals and unary predictions, i.e., $\mathcal{L} = \sum_i l(Q_i(v_i), p_\theta(v_i|O))$, where $l$ is the loss function selected according to the specific applications. In practice, we find the mean-square error of their logits works better for the tested datasets.

## J   SAMPLING STRATEGY

We sample mini-batches of ground formulae as in ExpressGNN Zhang et al. (2020). Specifically, in each optimization iteration, we sample a batch of ground formulae by randomly instantiating the arguments using entities in $C$. To make the inference results more reliable in each batch of training, the sampling is biased towards the facts with observations. Specifically, each grounding is sampled by gradually selecting the ground atoms. In selecting the ground atoms, we choose a ground atom in $O$ with the probability $p_g$. Otherwise, a random atom is grounded. This process is proposed in ExpressGNN and is proved to be effective in their results. This is reasonable since the estimation is very in-confident if all the ground atoms in the groundings are not observed, i.e., with rare evidence. In our approach, we also witness the effectiveness of the sampling approach and $p_g$ is also set to 0.9. We have investigated the performance of $p_g = 0$ and we list the results in Table 6. The results show that the sampling strategy is critical for performance. Nevertheless, the usage of the sampling strategy is rather a practical choice, considering the performance in UW-CSE/language is similar when $p_g = 0$.

## K   MODEL CAPACITY

In the compared methods, Zhang et al. (2020) also leverages a neural predictor, i.e., ExpressGNN, to learn the logical knowledge, similar to our neural predictor. The original paper showed that the use of ExpressGNN can effectively improve performance in their setting. However, in our experiments, we found our lightweight model with solely a list of embedding vectors is adequate and can achieve on-par results with a few parameters (embedding size = 128). Specifically, we showed the model parameter numbers in Table 7. Due to the varying number of constants, the number of parameters varies with different datasets. From the table, we can see that our model is more parameter-efficient than ExpressGNN. We have conducted experiments using ExpressGNN as the neural predictor and the results keep similar. Note that as the neural predictor can be implemented in parallel, various neural predictors make little difference to the training speed. These results indicate that the use of the ExpressGNN model is unnecessary in our work and therefore we use the simple embedding-based model instead.

## L   COMPARISON BETWEEN CLOSED-WORLD AND OPEN-WORLD ASSUMPTION

Our work follows the open-world assumption where all the unobserved facts are seen as latent variables. However, this assumption leads to severe scalability issues for many competitor methods. For

Table 7: The number of parameters of our neural predictor and ExpressGNN. "A., G., L., S., T." are abbreviations for "AI, Graphics, Language, Systems, Theory".

| | UW-CSE | | | | | Cora | | | | |
|---|---|---|---|---|---|---|---|---|---|---|
| | A. | G. | L. | S. | T. | S1 | S2 | S3 | S4 | S5 |
| ExpressGNN | 598K | 591K | 584K | 596K | 589K | 277K | 223K | 224K | 223K | 223K |
| Ours | 39K | 25K | 11K | 36K | 23K | 85K | 77K | 78K | 77K | 77K |

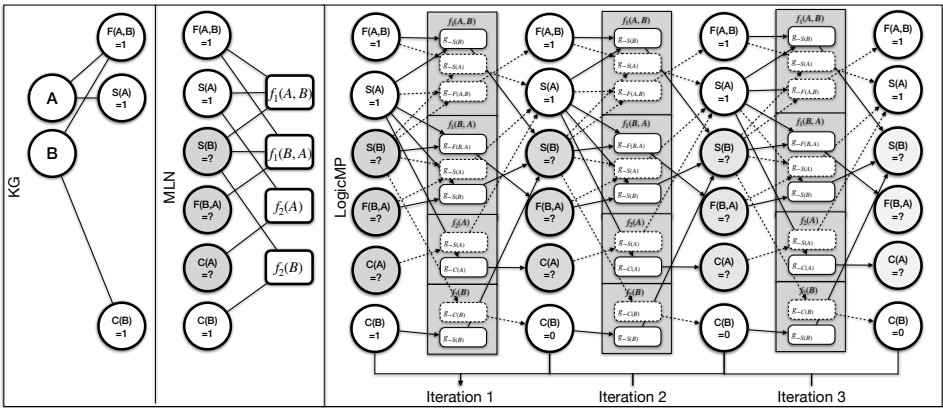

Figure 7: The illustration of the mean-field update by formulating the grounding messages as a set of hyper-edges in a graph. The computation can be achieved via scattering and gathering operations.

the UW-CSE and Cora datasets, the compared methods can hardly bear the computational complexity and fail to derive valid results. Therefore, we list the results under the closed-world assumption for these methods in the main paper, where unobserved facts (except the queries) are assumed to be false. We show that these methods typically perform better in the closed-world assumption as illustrated in Table 8. Even though, our method outperforms these methods by a considerable margin.

Table 8: AUC-PR of competitors under two assumptions on UW-CSE. Better results are in bold.

| Method | Assumption | UW-CSE | | | | |
|---|---|---|---|---|---|---|
| | | AI | Graphics | Language | Systems | Theory |
| MCMC | Open-world | - | - | - | - | - |
| | Closed-world | **0.19** | **0.04** | **0.03** | **0.15** | **0.08** |
| BP/Lifted BP | Open-world | 0.01 | 0.01 | **0.01** | 0.01 | 0.01 |
| | Closed-world | **0.21** | **0.04** | **0.01** | **0.14** | **0.05** |
| MC-SAT | Open-world | 0.03 | **0.05** | **0.06** | 0.02 | 0.02 |
| | Closed-world | **0.13** | 0.04 | 0.03 | **0.11** | **0.08** |
| HL-MRF | Open-world | 0.06 | 0.06 | 0.02 | 0.04 | 0.03 |
| | Closed-world | **0.26** | **0.18** | **0.06** | **0.27** | **0.19** |

## M    IMPLEMENTATION WITHOUT EINSTEIN SUMMATION

Without Einsum, we generate all grounding messages to create a graph with the hyper-edges. Fig. 7 shows an example of grounding messages. Since the number of latent variables varies in different groundings, we manually append some dummy variables to the batch of latent variables of $g_{-i}$. In the computation of the message, those dummy variables will be masked and replaced with the probabilities of 1 in the product of $Q_j(v_j = n_j)$. Note that there is an outer loop of formulae in Eq. 2 and we eliminate this by assigning the rule index to each grounding message. The model can be efficiently implemented using scattering and gathering operations. The overall complexity of an iteration is $\mathcal{O}(N^M L^2)$. Algorithm 2 illustrate the steps of iterations without Einsum where the messages are aggregated by gathering all messages of $G_f(i)$.

---

**Algorithm 2** The mean-field update w/o Einstein summation.

$Q_i(v_i) \leftarrow \frac{1}{Z_i} \exp(\phi_u(v_i)))$ for all $i$                                  $\triangleright$ Initialization

**for** $t \in \{1, ..., T\}$ **do**

    $\hat{Q}_{i,g}(v_i) \leftarrow \mathbf{1}_{v_i = \neg n_i} \prod_{j \in g_{-i}} Q_j(v_j = n_j)$                         $\triangleright$ Grounding Message

    $\hat{Q}_i(v_i) \leftarrow \sum_f w_f \sum_{g \in G_f(i)} \hat{Q}_{i,g}(v_i)$                              $\triangleright$ Aggregation

    $Q_i(v_i) \leftarrow \frac{1}{Z_i} \exp(\phi_u(v_i) + \hat{Q}_i(v_i))$             $\triangleright$ Adding unary potential & Normalization

**end for**

---

## N  MORE COMPARISON OF TRAINING EFFICIENCY

Table 9 illustrates the training velocity of groundings in a second as well as the total runtime in minutes with a batch size of 1024. These results show that LogicMP achieves around $10\times$ acceleration compared to ExpressGNN.

Table 9: The comparison of training efficiency and runtime.

| | Kinship/S1 | | UW-CSE/AI | | Cora/S1 | |
|---|---|---|---|---|---|---|
| method | ExpressGNN | LogicMP | ExpressGNN | LogicMP | ExpressGNN | LogicMP |
| groundings/second | 225 | 2,844 | 85 | 1,229 | 64 | 499 |
| runtime (minutes) | 80 | 8 | 4,024 | 276 | 5,333 | 684 |

## O  LIMITATION OF THIS WORK

A grounding in this paper and some previous work in the literature means an assignment to the arguments in the formulae or predicates. This implies that we are using the universal quantifier in the first-order logic. However, there is still an existential quantifier that is not considered.

The existential quantifier is more difficult than the universal quantifier. For a universal quantifier, any assignment leads to a ground formula that needs to satisfy. In contrast, the existential quantifier needs to consider multiple assignments jointly as it only requires there exists one assignment that satisfies the formula. We leave the study of existential quantifiers to future work. Currently, we feel that there are two plausible ways to tackle the problem. The first way is to replace the existential quantifier by a series of conjunctions, i.e., $\exists x, \mathtt{F}(x) \Leftrightarrow \vee_i \mathtt{F}(x_i)$. However, in some difficult situations, such replacement will construct an extremely complicated ground formula that is difficult to generalize. The second way is to transfer it to the Skolem norm form which can eliminate the existential quantifier by a map function that maps the existential variables to the universal variables. This is more promising as we can generalize the theorem to the existential quantifier via another mapping module like a special attention model. Since the tested benchmarks do not have any formulae with existential quantifiers, we leave the extension and verification to future work.

## P  ADDITIONAL RELATED WORK

**Symbolic reasoning.** The ability of symbolic reasoning is critical for intelligence (Winston, 1984) and is likely indispensable for nowadays AI research. A popular branch of symbolic learning is the logical specifications of neural networks (Dong et al., 2019; Reimann et al., 2022; Payani and Fekri, 2019; Marra and Kuzelka, 2021; Yang and Song, 2020) with constraints of logical knowledge (Badreddine et al., 2022; Riegel et al., 2020). Another research focus is the logic-based learning algorithms to enable the logical reasoning of common models (Hu et al., 2016; Xu et al., 2018; Guo et al., 2018; Xie et al., 2019). Besides, many methods were proposed to incorporate the symbolic reasoning into various tasks (Jiang and Luo, 2019; Li and Srikumar, 2019; Jiang et al., 2021). Our LogicMP can be seen as a specification of the Markov logic network to perform logical inference.

| smoke(a) | value |
|----------|-------|
| smoke(A) | 1 |
| smoke(B) | 0 |

| friend(a, b) | value |
|--------------|-------|
| friend(A, A) | 0 |
| friend(A, B) | 1 |
| friend(B, A) | 1 |
| friend(B, B) | 0 |

| smoke(a) | friend(a, b) | s(a) and f(a,b) |
|----------|--------------|-----------------|
| smoke(A) | friend(A, A) | 0 |
| smoke(A) | friend(A, B) | 1 |
| smoke(B) | friend(B, A) | 0 |
| smoke(B) | friend(B, B) | 0 |

Figure 8: (1) The ground atoms of the predicate "smoke". (2) The ground atoms of the predicate "friend". (3) The formula of "smoke(a) and friend(a, b)".

Table 10: Comparison between the results between neural predictor with and without LogicMP.

|  | UW-CSE | | | | | | Cora | | | | | |
|--|-----|-----|-----|-----|-----|------|-----|-----|-----|-----|-----|------|
|  | A. | G. | L. | S. | T. | avg. | S1 | S2 | S3 | S4 | S5 | avg. |
| neural predictor | .01 | .01 | .01 | .01 | .01 | .01 | .37 | .66 | .21 | .42 | .55 | .44 |
|  | (.00) | (.00) | (.00) | (.00) | (.00) | (.00) | (.03) | (.03) | (.01) | (.03) | (.03) | (.03) |
| neural predictor + LogicMP | .25 | .30 | .42 | .25 | .28 | .30 | .80 | .88 | .72 | .83 | .89 | .82 |
|  | (.02) | (.04) | (.03) | (.02) | (.05) | (.03) | (.01) | (.01) | (.01) | (.01) | (.00) | (.01) |

## Q    CONNECTION BETWEEN EINSUM AND PDB

Einsum is an efficient implementation of the summation of the product which can count the true groundings of conjunctive formulae. In the probabilistic database (PDB), the product can be achieved by the join operation and the summation can be seen as the aggregation operation. For example, for a conjunctive formula "smoke(a) and friend(a,b)", we can create a table with two columns $(x_1, x_2)$ where smoke(a) (friend(a, b)) corresponds to $x_1$ ($x_2$). The value of each column is the ground atom and the goal is to find the groundings that make the formula true.

In Fig. 8, (3) can be derived from (1) and (2). Note the "smoke(a) and friend(a, b)" is the premise of "smoke(a) and friend(a, b) $\rightarrow$ smoke(b)". Our mean-field method "counts" the groundings of each conjunctive premise that evaluate to true so that it can be implemented efficiently via parallel Einsum operation. Note that "counts" is intuitive as $Q$ is a continuous probability rather than discrete 0/1.

## R    ADDITIONAL RESULTS OF NEURAL PREDICTOR

We attach the results using the neural predictor only in Table 10. The experimental results show that the neural predictor only without LogicMP performs poorly in both the UW-CSE and Cora datasets. As expected, without the explicit use of logical knowledge, the neural predictor can hardly learn useful patterns for the generalization in the tested queries. Note that since UW-CSE has no negative samples (i.e., false facts), we create negative samples for class balance by treating unobserved facts as false, and half samples in the batch are drawn from these negative samples.

