# OpenReview forum: "An Efficient Mean-field Approach to High-Order Markov Logic"
_ICLR.cc/2023/Conference — Submitted to ICLR 2023_

### Official Review · Reviewer_qmoi · 2022-10-17

**Confidence:** 3
**Correctness:** 2
**Technical Novelty And Significance:** 3
**Empirical Novelty And Significance:** 2
**Recommendation:** 3

**Clarity, Quality, Novelty And Reproducibility:**

I found the paper very hard to read in multiple sections, as well as a number of unsubstantiated claims.

---

    "[...] lifted methods are not only difficult to implement ..."

Why?

    "... but also infeasible when the symmetric structure becomes invalid, e.g., one other model is integrated."

What does 'one other model is integrated' means in this context? Also, there exists work on lifted inference in asymmetric MLNs [2].

---

The role of latent variables in Eq.1 is not clear to me. This is also the first time that I encounter LVs in the context of MLNs. Somewhere in the text it is mentioned that latent variables are used to model unobserved facts, under an open world assumption (OWA). Yet, most works on MLNs use CWA. I feel like the OWA made in this work is crucial for understanding the whole approach, but it is only mentioned in the experiments (Sec. 5.2) and its discussion is deferred to Appendix L.

---

How mean-field variational inference can be used in MLN is also unclear. Adding an example to Section 4 would be beneficial as Fig. 1 is not really informative in this regard.
How is the variational distribution constructed? What parameters are optimized to minimize the KL divergence?
The work supposedly uses neural networks, but it is not clear how these LogicMP layers are structured or trained. What are the parameters of LogicMP?

---

I also found the use of Einsum optimization quite convoluted.

    "Naturally, we can generate all the propositional groundings in Gf as a set of hyper-edges to perform the aggregation."

Hyper-edges over what? No hyper-graph is mentioned before this sentence.

The abstract claims that "In most practical cases, it can reduce the complexity significantly to polynomial for the formulae in conjunctive normal form (CNF)".
This is a very strong statement. What 'most cases' means exactly? I couldn't find a formal definition of this set of cases in the text.

---

The experimental section gives some clues on the proposed approach,
but also raises more questions.

    "In general, LogicMP poses no constraints to its applications and can incorporate logic rules for various purposes."

This is a very vague sentence. I thought that LogicMP layers are used to approximate inference in MLNs.

    "Ideally, it can serve as a logic CRF for arbitrary neural networks and be trained end-to-end via maximum likelihood estimation (MLE) as in CRFasRNN (Zheng et al., 2015). However, as the labeled data is rare in our datasets, MLE is prone to overfitting and we turn to semi-supervised learning where rules act as priors to infer the latent variables."

This paragraph is hinting at different applications.

    "We fix the formula weights of 1 [...] for all the experiments."

Why? Shouldn't the weights of the MLN be learned from data? If these weights are fixed, what is optimized during the training?

    "Note that LogicMP can perform full graph computation efficiently, but the practical results indicate that training with sampling is more stable for the learning of logical knowledge for the backbone model."

What does 'full graph computation' mean? What is the 'backbone model'? The notion was never mentioned before.

---

Notes:

Hinge-Loss Markov Random Fields (HL-MRFs) is not an inference technique for MLNs, but a different (albeit similar) probabilistic model. See Section 4.2 in [1].

References:

[1] Bach, Stephen H., et al. "Hinge-loss markov random fields and
probabilistic soft logic." (2017).

[2] Van den Broeck, Guy, and Mathias Niepert. "Lifted probabilistic
inference for asymmetric graphical models." AAAI, 2015.


**Strength And Weaknesses:**

Pros:
+ MLNs represent a very popular class of relational probabilistic models.
+ Advances in MLN inference are valuable for a broad audience.

Cons:
- In my opinion, the presentation of the proposed approach should be greatly improved before publication (details in the Clarity section).

**Summary Of The Paper:**

This paper proposes an approximate inference algorithm for Markov Logic networks (MLNs) with disjunctive clauses.  The exponential grounding of the first-order structure is mitigated by leveraging a neural network architecture called LogicMP, which implements an efficient mean field approximation of the true posterior.


**Summary Of The Review:**

The problem considered in this work is relevant, but I had an hard time reading the paper. I fully read the paper multiple times, but there are still important aspects of the proposed approach that I don't understand. I cannot reccomend the current version for publication.

---

> ### Author Response · Authors · 2022-11-16
> **Response to Reviewer qmoi (PART 1)**
>
> Thanks very much for your effort in reading our paper. Due to the main paper limit, we have to simplify some descriptions and move less important details to the appendix. We have uploaded a new version where the modifications are highlighted in red color according to your suggestions and comments.
>
> To summarize our work, we propose to use the mean-field algorithm for MLN inference, which is a long-standing problem. We show that the mean-field update of MLN can be formulated as the forward tensor computation that can be implemented by LogicMP. Formulating joint inference as a neural network enables efficient implementation and broad usage across various neural architectures. We list detailed responses as follows.
>
> > Comment: Use open-world assumption?
>
> Yes, we use open-world assumption and the latent variable denotes the unobserved fact. Thanks for pointing it out and we add a sentence at the beginning of Sec. 3.1 in the new version -- "Note that we consider all unobserved facts as the latent variables to infer under the open-word assumption."
>
> > Comment: How mean-field algorithm is applied to MLN? What is the variational distribution? What parameters are optimized to minimize KL? What are the parameters of LogicMP?
>
>
> The prior distribution $p(\mathbf{v}|O)$ is an MLN. Since $p(\mathbf{v}|O)$ is generally intractable, we use the mean-field algorithm for approximate posterior marginal inference. Specifically, the MF algorithm computes a **variational distribution** $Q(\mathbf{v})$ that best approaches $p(\mathbf{v}|O)$ where $Q(\mathbf{v})=\prod_{i} Q_i(v_i)$ is a product of independent marginal distributions over each latent variables. Note that $Q_i(v_i=0)$ and $Q_i(v_i=1)$ are two scalars that form a categorical distribution for inference. The algorithm minimizes the KL divergence $D_{KL}(Q(\mathbf{v})||p(\mathbf{v}|O))$ (note that minimizing $D_{KL}(Q(\mathbf{v})||p(\mathbf{v}|O))$ is equivalent to maximizing the evidence lower bound (ELBO) of $\log p(O)$). By considering $D_{KL}(Q(\mathbf{v})||p(\mathbf{v}|O))$ as a function of $Q_i(v_i)$, [1,2] shows that we can optimize $Q_i$ and the optimal $Q_i$ can be derived in closed-form (cf. Sec. 2.4 in [1] for the details of the mean-field algorithm and [2]). We have revised Sec. 3.1 and add the detailed derivation of the update equation in Appendix A to show how the mean-field algorithm is applied to MLN.
>
> The derived closed-form solution is the update equation in Eq. 2 and 3. In contrast to the variational EM where the posterior model with a set of additional tunable parameters is optimized to minimize the ELBO, the mean-field update equation estimates $Q_i(v_i)$ in **closed-form** which is parameter-free and purely configured by MLN.
>
> Since LogicMP is simply the implementation of the mean-field update that is derived from MLN, the model parameter of LogicMP only contains the rule weights, i.e., $w_f$, which are fixed at 1 without learning in the current work.
>
> [1] Blei David M., et al. "Variational inference: A review for statisticians." Journal of the American Statistical Association 2017.
> [2] https://www.cs.princeton.edu/courses/archive/fall11/cos597C/lectures/variational-inference-i.pdf
>
> > Comment: Adding an example to Section 4 would be beneficial as Fig. 1 is not really informative in this regard.
>
> In the revised version, we illustrate how the grounding message is computed in the framework figure (Fig. 1) with the corresponding caption
>
> "The calculation of the message w.r.t. a single grounding is illustrated in the "grounding message'' block with dashed lines: the message from $\mathtt{S}(A)$ and $\mathtt{F}(A,B)$ w.r.t. $f_1(A,B)$ can be simplified as $Q_{\mathtt{S}(A)}(1)Q_{\mathtt{F}(A,B)}(1)$."
>
> , and added the example in Sec. 4.1
>
> "Consequently, the grounding message can be simplified, e.g., $Q_{\mathtt{S}(A)}(1)Q_{\mathtt{F}(A,B)}(1)$."
>
>
> > Comment: The notation of hyper-edge is confusing.
>
> Thanks for pointing it out and we have removed this notation. The sentence in Sec. 4.2 becomes ''Naturally, we can generate all the propositional groundings in $G_f$  to perform the aggregation.". The hyper-edge in the literature typically denotes the edge linking a set of nodes. Since the grounding message is from $g_{-i}$ with multiple variables to a single variable $g_i$, we borrow the notation for convenience. The illustration of the graph can be found in Appendix M and Fig. 7.

---

> ### Author Response · Authors · 2022-11-16
> **Response to Reviewer qmoi (PART 2)**
>
> > Comment: What does "most cases" exactly mean?
>
> The optimal optimization of Einsum is NP-hard [1]. By using a search algorithm, the final complexity depends on the format of the formula and the order of arguments in the contraction (cf. last paragraph in Sec. 4.2). Therefore, we cannot give a specific scope of formulae that can be optimized. As a remedy, we emphasize  "**practical**" around "most cases" statements throughout the paper trying to avoid overclaiming. In practice, the widely-used formulae are typically the chain rules and chain-like compositional rules that can be optimized effectively. To enhance intuitive understanding, we give several examples of optimized einsum in Appendix G. An example of a formula that can not be optimized is "R1(a, b, c, d) and R2(b, c) and R3(c, d) and R4(a, d) -> R0(a, c)" (and "R1(a, b, c) and R2(b, c, d) and R3(c, d) and R4(a, d) -> R0(a, c)").
>
> Note: Despite NP-hard, the optimization for einsum is negligible in practice for the used formulae with <= 5 arguments via the opt_einsum library.
>
> [1] Chi-Chung, et al. On optimizing a class of multi-dimensional loops with reduction for parallel execution. Parallel Processing Letters 1997.
>
> > Comment: "In general, LogicMP poses no constraints to its applications and can incorporate logic rules for various purposes." The sentence is very vague. I thought that LogicMP layers are used to approximate inference in MLNs.
>
> Yes, LogicMP is used to perform the approximate MLN inference. By "various purposes", we mean the learning settings such as supervised and semi-supervised learning as explained later in that paragraph. This sentence is rewritten as "In general, LogicMP performs MLN inference and can incorporate logic rules in various learning settings".
>
> > Comment: How LogicMP is used in the experiments? why the formula weight is fixed? What is the backbone model?
>
> In our approach, LogicMP acts as an MLN regularizer for a **backbone** neural predictor (cf. model details in Sec. 5.1 and the details in Appendix I). LogicMP layers are used to approximate inference in MLNs. The neural predictor ("backbone") learns to fit the inference results from the LogicMP.
>
> The inference part (LogicMP) is fixed and the rule weights are fixed as 1, while the backbone part (neural predictor) has parameters to optimize.
>
> To avoid misunderstanding, we unify the reference to the backbone neural network as "neural predictor" and revise the sentence to "... for the learning of logical knowledge for the neural predictor".
>
> > Comment: What does "full-graph computation" means?
>
> The full-graph computation is against the sampling strategy used in our approach (cf. model details in Sec 5.1 and Appendix J). By "sampling", we mean sampling large batches of groundings for stochastic training. Appendix J demonstrates how the sampling is performed and why it is preferred.
>
> > Comment: Why "these lifted methods are not only difficult to implement but also infeasible when the symmetric structure becomes invalid, e.g., one other model is integrated."?
>
> To be clear, we now revise the sentence to "these lifted methods typically become infeasible when the symmetric structure breaks down, e.g., unique evidence is integrated for each variable."
>
> Previous lifted methods have studied the problem of MLN inference extensively which give us a lot of inspiration, however, there still is room to improve. As stated in [2], lifted variable elimination [1] is "extremely complex, generally does not scale to realistic domains, and has only been applied to very small artificial problems". Lifted belief propagation [2] constructs the minimal lifted network via merging the nodes. In the worst case, the lifted network can have the same size as the original ground network. Lifted MCMC [3] leverages the symmetric structure to perform sampling in the Markov chain, but its scalability in the practical datasets is not fully verified in their work.
>
> By "one other model is integrated", we mean the use of unique unary potentials $\phi_u$ for each distinct variable as the soft evidence. [3] mentioned that "typical lifted inference methods break down when the symmetry is removed, e.g., soft evidence is provided". [4] leverage a mixed Markov chain to perform lifted sampling in an asymmetric graph model with a symmetric structure. Our work naturally adapts to the case with soft evidence, using an entirely different mean-field methodology. Our work is a parallel method to such asymmetric graph models.
>
> [1] Rodrigo de Salvo Braz, et al. "Lifted First-Order Probabilistic Inference". IJCAI 2005.
> [2] Parag Singla, et al. "Lifted First-Order Belief Propagation". AAAI 2008.
> [3] Mathias Niepert, et al. "Lifted Probabilistic Inference: An MCMC Perspective". StarAI@UAI 2012.
> [4] Van den Broeck, et al. "Lifted probabilistic inference for asymmetric graphical models." AAAI, 2015.

---

### Official Review · Reviewer_gHdR · 2022-10-24

**Confidence:** 4
**Correctness:** 4
**Technical Novelty And Significance:** 3
**Empirical Novelty And Significance:** 3
**Recommendation:** 6

**Clarity, Quality, Novelty And Reproducibility:**

The paper is clearly written and it reports a new method to do efficient inference in MLNs

**Strength And Weaknesses:**

+ Very interesting idea
+ Well written paper
- lack of experimental results


**Summary Of The Paper:**

The paper proposes a new method for inference in MLNs.


**Summary Of The Review:**

The paper is well written and the claims are supported by theoretical proofs.

It is not clear whether the structure of the MLNs are fixed or learned from scratch. In this case this should be explained in the paper.
In particular, while the part concerning the mean-field approach is very clear and correctly presented, there is a lack of description about the structure learning adopted in the paper.

Furthermore, more datasets could be used in order to prove the validity of the approach.

Finally, the paper is interesting but could be improved integrating some parts as reported above.

---

> ### Author Response · Authors · 2022-11-16
> **Response to reviewer gHdR**
>
> Thanks very much for your interest in our work and your helpful comments.
>
> > Comment: Does our method learn the structure of MLNs?
>
> No, since this work focuses on the inference problem of MLN, the structure of MLN is fixed. The structure of the probabilistic graph of MLN is defined by the formulae and the knowledge base. In our work, the formulae are given and fixed,  and no other formulae will be induced during the computation. We also fix the rule weight to 1 in the experiments (cf. Sec. 5.1). The structure learning of LogicMP is very important and we want to investigate it in the future. We add a sentence "This paper focuses on the inference problem of MLN with the fixed MLN structure." at the beginning of Sec. 3.1. to make it clear.
>
> > Comment: Lack of experimental results.
>
> This paper proposes an efficient mean-field approach for MLN inference. In the experiments, we aim to show (1) the effectiveness (performance) of LogicMP (2) the effect of multiple LogicMP layers, and (3) the efficiency of LogicMP. We have conducted extensive experiments to verify these perspectives and each experimental score is the average of 5 runs.
>
> Specifically, the tested four datasets are widely used benchmarks for symbolic reasoning models. The smoker dataset is an artificial dataset for sanity checking and the experimental results in the kinship dataset show the effectiveness in precise reasoning. The UW-CSE and Cora datasets are the standard benchmarks for symbolic reasoning since the MLN was proposed in 2006. Despite a decade of improvement, these two benchmark datasets remain challenging. Traditional methods can hardly perform inference in UW-CSE under the open-world assumption where all unobserved facts are seen as latent variables. Even with the recent variational EM method, the AUC-PR is at the level of 20%. In contrast, LogicMP can achieve an average AUC-PR of 30% and the improvements are consistent across five splits. The Cora is a relatively large dataset for MLN with more than 140K ground atoms and 300 billion ground formulae. In such a dataset, LogicMP can still outperform several existing MLN-based inference algorithms as well as ExpressGNN by an evident margin (from 68% to 82%).
>
> Besides, we also conducted extensive experiments to ablate the number of LogicMP layers in Sec. 5.3 and show the training efficiency in Sec. 5.4. The ablation study is performed in all splits of three datasets to draw a practical conclusion and the efficiency is compared with quantitive results. These experimental results are effective to analyze the effect and show the efficiency of the proposed method.
>
> We are currently working on the extension of LogicMP on other tasks like the NLP event extraction task to incorporate logical knowledge using end-to-end training. We have obtained some preliminary results and leaving them to another work is better for clear representation.

---

### Official Review · Reviewer_cRow · 2022-10-25

**Confidence:** 4
**Correctness:** 3
**Technical Novelty And Significance:** 2
**Empirical Novelty And Significance:** 2
**Recommendation:** 5

**Clarity, Quality, Novelty And Reproducibility:**

The paper is mostly well written and has rigorous complexity analysis. The significance of the work could have been demonstrated better wth relation to prior work as well as experiments with more state-of-the-art systems.

**Strength And Weaknesses:**

Strengths
-The use of Mean-Field inference for MLNs through deep model layers seems like a novel contribution to me.
-The complexity analysis is performed rigorously and shows the scalability of the proposed method.
-Experiments show that the proposed method scales well over large number of groundings while also being accurate

Weakness
-One weakness is that the problem of controlling the exponential grounding problem in MLNs has been explored in earlier work. For example, Venugopal et al. (AAAI 2015) model the same problem using a junction tree formulation and use approximate message passing to scale up the computations. In such cases, the complexity is dependent not on the number of groundings but the width of the tree-decomposition. For a chain formula, this width is small but for more densely connected formulas, this is large. Other similar approaches have also been explored.
Thus, I am not sure how the proposed method compares in terms of significance.

-The experiment baselines particularly for CORA does not seem as strong. Another approach that implements MLNs through deep models (Graph Markov Neural Nets, Qu et al. ICML 2020) seems to show better performance. In general, I feel experiments on just two datasets (since smokers is more of a toy dataset) does not strongly demonstrate the significance of the proposed method.


**Summary Of The Paper:**

The paper proposes to improve inference in Markov Logic Networks through the use of deep models. In particular, Mean Field inference is implemented as layers which pass messages to compute summations more efficiently by exploiting logical structure in the MLN.

The time complexity analysis is performed to show that the exponential complexity of a Mean Field iteration is reduced for chain ruled formulas in the MLN. Further by parallelizing, we can achieve far salable computations of the messages in the MF iterations. Einstein summation is used to reduce complexity of enumerating all possible groundings of a first-order formula. It is shown that in some cases, even when the total number of groundings of the formula is large, the complexity of computing messages can be reduced by performing summations using Einsum.

Experiments are performed with 3 datasets in the MLN literature. Comparisons are with existing MLN-based inference algorithms as well as ExpressGNN which is a GNN-based approach for MLN inference. Results indicate that the proposed approach is accurate and also scale well with increased number of groundings.


**Summary Of The Review:**

The connection of Mean Field iterations and the deep model layers seems nice. Also, parallelizing the computations and demonstrating scalability is a nice contribution since inference in MLNs is known to be hard due to the large number of groundings. However, as I understand it, other approaches have tried to address the grounding problem in MLNs, so maybe the novelty is limited.

---

> ### Author Response · Authors · 2022-11-16
> **Response to Reviewer cRow**
>
> Thanks very much for your invaluable time and helpful comments.
>
> > Comment: Controlling the exponential grounding problem has been explored in the earlier work.
>
> The acceleration of the counting for MLN is an important problem, which gives rise to many methods, such as cited [1] and mentioned [2,3,4]. They typically convert the counting problem into a #SAT problem and leverage the related techniques to reduce the complexity. Our method distinguishes among those methods in several aspects:
>
> ‒	**Our method uses einsum implementation that is more efficient than previous methods.** [2] needs to perform message passing over the junction tree while LogicMP explicitly uses einsum for parallel computation. [3] noted that [2] is practically inefficient -- ''the number of entities can be quite large (thousands), even the treewidth of 3 may be infeasible''. In contrast, our method uses a GPU operation that easily scales to thousands of entities. The methods like [3] are approximate approaches but einsum is precise. Other approaches like [4] use other tools such as the database but none of them explicitly proposes to use einsum, let alone its optimization for complexity reduction.
> ‒	**LogicMP is theoretically derived from the mean-field algorithm, which is in principle different from previous methods.** It sums the true conjunctive premises with continuous marginals, rather than discrete 0/1, and it straightforwardly generalizes to CNF formulae in the context of the mean-field approach. The inference with CNF can be decomposed into clauses by Theorem 4.3.
> ‒	**LogicMP implements the MLN inference by a set of forward tensor computations which bridges the gap between the MLN inference and neural networks.** The use of einsum and its optimization enable efficient computation so that it can be easily integrated as a feed-forward module.
>
> In the revision, we add the citations in the related work - "Despite the effort in improving the efficiency [...,2,3], the MLNs still struggle in efficient inference".
>
> [1] Parag S. et al. Memory-Efficient Inference in Relational Domains. AAAI 2006.
> [2] Venugopal D, et al. Just count the satisfied groundings: Scalable local-search and sampling-based inference in mlns. AAAI 2015.
> [3] Sarkhel S, et al. Scalable training of Markov logic networks using approximate counting. AAAI 2016.
> [4] Das M, et al. Scaling lifted probabilistic inference and learning via graph databases. ICDM, 2016.
>
> > Comment: ... baselines particularly for CORA do not seem as strong... Graph Markov Neural Nets (ICML 2020) seems to show better performance.
>
> The Cora dataset used in our paper is not the Cora used in the Graph Markov Neural Network (GMNN) and the results are not directly comparable. It may be confusing since several "Cora" datasets are available on the websites. We added a footnote in Sec. 5.1 of the revised version -- "This is not the Cora dataset (Sen et al. (2008)) typically for the graph node classification."
>
> - [G-CORA](https://github.com/DeepGraphLearning/GMNN/tree/master/semisupervised/data/cora): Cora used in GMNN is the one for the node classification tasks on the graph [1]. It is typically used to verify the capacity of representation learning of GNNs, such as GATs. It has node features and edges for representation learning but no rule for explicit symbolic reasoning.
> - [L-CORA](https://github.com/expressGNN/ExpressGNN/tree/master/data/cora/S1): The Cora dataset used in LogicMP is proposed in the early MLN paper [2]. It is in the standard format of symbolic reasoning with specific facts, predicates, and rules. We give a comparison between them as follows.
>
> |dataset|#predicate| #provided rules|node feature|logical knowledge|common usage|
> |-|-|-|-|-|-|
> |G-CORA|1|0| Yes| No|representation learning for deep GNNs. |
> |L-CORA|10| 32| No| Yes| symbolic reasoning for MLN-related models. |
>
>
> GMNN combines GNN with MRF by minimizing $KL(q||p)$ via variational EM. Note that GMNN does not explicitly model the logic in the score function as both $q$ and $p$ are modeled by ''black-box" GNNs. Strictly speaking, GMNN can not be seen as an MLN-based model as it lacks an explicit combination of logical knowledge. It is more like GNN + MRF rather than GNN + MLN.
>
> Table 2 of GMNN has a result of "MLN" for the G-CORA. By Sec. 6.2 in GMNN paper -- "In MLN, we simply use one indicator function in the potential function... judges whether the objects in a clique have the same label", as G-CORA has no rule, they heuristically create a hand-craft rule -- "the linked nodes have same labels". Solely one hand-craft rule may be inadequate to verify the ability of symbolic reasoning. We feel that G-CORA is not a perfect benchmark to test MLN-related methods. In contrast, L-CORA has 32 precise rules and the inference in L-CORA is still challenging.
>
> [1] Sen P, et al. Collective classification in network data. AI magazine, 2008.
> [2] Parag Singla, et al. Discriminative training of Markov logic networks. AAAI 2005.

---

### Official Review · Reviewer_sKHS · 2022-10-27

**Confidence:** 3
**Correctness:** 4
**Technical Novelty And Significance:** 3
**Empirical Novelty And Significance:** 4
**Recommendation:** 6

**Clarity, Quality, Novelty And Reproducibility:**

The clarity of the paper is good, although the notations are quite dense.

The idea of performing mean field inference on MLN using einsum is interesting and author further demonstrated its applications in various
prediction tasks.

On the reproducibility, author explained well on the implementation of LogicMP layer, and they also explain how the LogicMP layer interplay with a neural network backbone for generating predictions in relational problems.

I have two questions/comments:

1. It is known that probabilistic database (PDB) can be converted to a MLN [1]. There is an inference algorithm in PDB that uses relational join and aggregation operator, which is similar to the einsum. The inference algorithm is exact for safe query, and can be an approximation when the query is not safe. Is there any possible connections between the proposed method and PDB inference literature.
[1] https://simons.berkeley.edu/sites/default/files/docs/5662/talk-simons-2016.pdf

2. On the  experiment result shown in Table 2, can author also present the quality of the NN backbone alone without the distillation from the LogicMP? I think this could be a better motivation for neural modelers of applying LogicMP when there are soft logical connections between the outputs.



**Strength And Weaknesses:**

Strength:
1. It is likely the first work that applies mean field on MLNs. The reduction of the mean field procedure using einsum makes the procedure easy to be implemented and accelerated.
2. Author shows good result of using MLNs as a regularizer for learning a model in relational application, e.g. UW-CSE and Cora.

Cons:
1. The notation introduced in the paper is a little bit dense, e.g. variables with many super scripts and lower scripts. It was challenging for the reviewer to go through the equations within few passes. One unsolicited suggestion is to remove some of the super script when the context is clear. E.g. In Theorem 4.3, the super script on the n might be known from the context and can be dropped. Alternatively, adding more examples on how each equation is computed might also help.

**Summary Of The Paper:**

The paper proposed using tensor operator, i.e. einsum, to perform mean field inference in MLN. Author demonstrated that each iteration of the MF message passing can be efficiently computed using a sequence of einsum operation. This not only enables one to perform approximate inference in MLN, the inference procedure can easily implemented as neural layers to perform neural-symbolic learning. In particular, author demonstrated that it achieves good results on relational prediction tasks when combining a neural predictor and a MLN regularizer.

**Summary Of The Review:**

In general, the paper delivers an interesting approach to perform inference in MLNs, and authors further demonstrated that the inference procedure can be integrated with a neural backbone to perform prediction tasks while leveraging the logical connections between the output.

---

> ### Author Response · Authors · 2022-11-16
> **Response to Reviewer sKHS**
>
> First of all, thanks very much for your effort in understanding our work and your positive feedback.
>
> > Comment: The notation introduced in the paper is a little bit dense. Remove some of the superscript when the context is clear. Alternatively, adding more examples on how each equation is computed might also help.
>
> Thanks for the suggestion. In the revised version, we have removed the superscript when the context is clear. We also illustrated how the grounding message is computed in the framework figure with the corresponding caption
>
> “The calculation of the message w.r.t. a single grounding is illustrated in the ``grounding message'' block with dashed lines: the message from $\mathtt{S}(A)$ and $\mathtt{F}(A,B)$ w.r.t. $f_1(A,B)$ can be simplified as $Q_{\mathtt{S}(A)}(1)Q_{\mathtt{F}(A,B)}(1)$.”
>
> , and added an example in Sec. 4.1
>
> "Consequently, the grounding message can be simplified, e.g., $Q_{\mathtt{S}(A)}(1)Q_{\mathtt{F}(A, B)}(1)$."
>
>
> > Comment: Connection between the relational join and aggregation operation in PDB and Einsum.
>
> Einsum is a method to perform the summation of the product which is an analogy to the model counting problem in PDB. Even though, einsum, differently from general PDB which enumerates assignments to the ground atoms, enumerates over groundings. It also enables efficient implementation for GPU in the mean-field MLN inference. We state such a connection in Appendix Q.
>
> Specifically, Einsum is an efficient implementation of the summation of the product which can count the true groundings of conjunctive formulae. In the PDB, the product can be achieved by the join operation and the summation can be seen as the aggregation operation. For example, for a conjunctive formula ``smoke(a) and friend(a,b)'’, we can create a table with two columns ($x_1$, $x_2$) where smoke(a) (friend(a, b)) corresponds to $x_1$ ($x_2$). The value of each column is the ground atom and the goal is to find the groundings that make the formula true. The following tables illustrate the analogy.
>
> Table 1: The ground atoms of the predicate ''smoke''.
>
> | smoke(a) | value |
> |---|---|
> | smoke(A) | 1 |
> | smoke(B) | 0 |
>
> Table 2: The ground atoms of the predicate ''friend''.
>
> |friend(a, b) | value |
> |---|---|
> |friend(A, A) | 0|
> |friend(A, B) | 1|
> |friend(B, A) | 1|
> |friend(B, B) | 0|
>
> Table 3: The formula of "smoke(a) and friend(a, b)".
>
> | smoke(a) | friend(a, b) | smoke(a) and friend(a,b) |
> |-|-|-|
> | smoke(A) | friend(A, A) | 0|
> | smoke(A) | friend(A, B) | 1|
> | smoke(B) | friend(B, A) | 0|
> | smoke(B) | friend(B, B) | 0|
>
>
> Table 3 can be derived from Tables 1 and 2. Note that "smoke(a) and friend(a, b)" is the premise of "smoke(a) and friend(a, b) -> smoke(b)". Our mean-field method "counts" the groundings of each conjunctive premise that evaluate to true so that it can be implemented efficiently via parallel Einsum operation. Note that "counts" is intuitive as $Q$ is a continuous probability rather than discrete 0/1.
>
> > Comment: Show the quality of the NN backbone alone without the distillation from the LogicMP and use it as a motivation for the neural modelers.
>
> The experimental results show that the neural predictor only without LogicMP performs poorly in both the UW-CSE and Cora datasets. We attach the results in Appendix R in the revision.
>
> Here we show the results in the following tables. As expected, without the explicit use of logical knowledge, the neural predictor can hardly learn useful patterns for the generalization in the tested queries. Note that since UW-CSE has no negative samples (i.e., false facts), we create negative samples for class balance by treating unobserved facts as false, and half samples in the batch are drawn from these negative samples.
>
> Table 4: Results of the neural predictor with and without LogicMP on the UW-CSE dataset. The bracket denotes the standard deviation.
> |  | uw_cse/ai | uw_cse/language | uw_cse/graphics | uw_cse/systems | uw_cse/theory | uw_cse/avg |
> |-|-|-|-|-|-|-|
> | neural predictor  | .01 | .01  | .01  | .01  | .01 | .01 |
> |  | (.00)  | (.00)  | (.00)  | (.00)  | (.00) | (.00) |
> | neural predictor + LogicMP | .25  | .30  | .42  | .25 | .28 | .30 |
> |  | (.02)  | (.04)  | (.03) | (.02)  | (.05)   | (.03)  |
>
> Table 5: Results of the neural predictor with and without LogicMP on the Cora dataset. The bracket denotes the standard deviation.
> |   | cora/S1 | cora/S2 | cora/S3 | cora/S4 | cora/S5 | cora/avg |
> |-|-|-|-|-|-|-|
> | neural predictor  | 0.37    | 0.66    | 0.21    | 0.42    | 0.55    | .44      |
> |   | (0.03)  | (0.03)  | (0.01)  | (0.03)  | (0.03)  | (0.03)   |
> | neural predictor + LogicMP | .80     | .88     | .72     | .83     | .89     | .82      |
> |  | (.01)   | (.01)   | (.01)   | (.01)   | (.00)   | (.01)    |

---

> > ### Comment · Reviewer_sKHS · 2022-11-28
> > **Thank you for the response!**
> >
> > Thank you for the response.
> >
> > Thanks for sharing the empirical results that compares between neural predictor and neural predictor + LogicMP! It looks great!
> >
> > I still have question on the comments regarding to the connection with OpenDB.
> >
> > I am looking at p69 of [1]. It seems that one can convert a soft constraint from MLN into a relation in PDB where each column represent a variable from the original formula. From the example in [1], the generated relation A has columns x and y instead of propositional relation R(x,y) and S(y). Based on that transformation, the inference of PDB is also performing some tensor einsum operation where each tensor have axes whose scopes cover all the constants from the MLN/PDB.  I was wondering whether the PDB safe query inference algorithm is computing approximate solution that is similar to the mean field approximation suggested by this paper.
> >
> > I am not experienced with lifted inference literature, hence my intuition might be incorrect. I also noticed the example shown on p69 of [1] only contains one soft formula. I am not sure whether the conversion could be applied to arbitrary MLNs either.
> >
> > [1] https://simons.berkeley.edu/sites/default/files/docs/5662/talk-simons-2016.pdf

---

> > > ### Author Response · Authors · 2022-12-02
> > > **Discussion of connections between PDB and LogicMP**
> > >
> > >
> > > Thank you very much for your reply.
> > >
> > > The PDB safe query inference algorithm is different from our mean-field algorithm mainly because the PDB safe query inference corresponds to a weighted first-order **model counting problem** over the worlds (i.e., the assignments to the ground atoms) but the einsum in LogicMP can be intuitively seen as the **counting problem** over a single world with marginal probabilities of the ground atoms. The model counting problem that enumerates the worlds is more difficult than the counting problem that enumerates the groundings in the single world. By the variational mean-field algorithm, we can mitigate the problem of counting massive worlds, which is exponential in the number of ground atoms.
> > >
> > > Note: In the model counting problem, a model = an assignment to the ground atoms in the MLN literature = a world in the MLN/PDB literature.
> > >
> > > The PDB safe inference tackles the weighted first-order model counting problem. For the safe queries, it can operate the tensors/tables on the axes to perform the lifted inference to integrate the worlds. In contrast, the einsum in our variational approach that operates tensors for CNF can be intuitively regarded as estimating the expected counting over the single world with the marginal probabilities of the ground atoms. We summarize these two approaches in Table 1.
> > >
> > > Table 1: Comparison between the safe inference algorithm of PDB and the mean-field approximation.
> > >
> > > |  | corresponding problem  | counting scope | valid formula|
> > > |---|---|---|---|
> > > | Safe inference in PDB |weighted first-order model counting | all worlds | safe queries (p19-23 [2])|
> > > | Einsum of LogicMP in MLN |variational approximation of counting problem | all groundings in a single world | CNF |
> > >
> > >
> > > [1] provides a more detailed connection between MLN and PDB -- both can be formulated as the weighted first-order model counting problem. As stated in p137 of [1] and p69 of [2], we can construct a new proxy relation for a formula in MLN (cf. Table 2). By this transformation, we can formulate an MLN into a weighted first-order model counting problem (cf. p121 in [1]). However, this transformation is just used to show the equivalence. To obtain the weight of a certain world, we face another problem -- the counting problem which counts the true ground formulae. With a certain world, the weight of the world is calculated by retrieving the values in the relation tables (weight for MLN and probability for PDB). Retrieving table values w.r.t. a certain world from Table 2 to obtain the true ground formulae in Table 3 (i.e., the specific tuples in the weight function of p69 [2]) corresponds to the counting problem. The einsum of our approach is the variational approximation of this counting problem.
> > >
> > >
> > >
> > > Table 2: New relation A for the MLN formula R(x, y) -> S(y) with arguments x,y where w4 for the true ground formula and w4' for the false ground formula. X=(a1, a2, a3), Y=(b1, b2).
> > >
> > > | x   | y   | w(F(x, y)) | w(not F(x, y)) |
> > > |-----|-----|------------|----------------|
> > > | a1   | b1   | w4         | w4'            |
> > > | a1   | b2   | w4         | w4'            |
> > > | a2   | b1   | w4         | w4'            |
> > > | a2   | b2   | w4         | w4'            |
> > > | a3   | b1   | w4         | w4’          |
> > > | a3   | b2   | w4         | w4’          |
> > >
> > >
> > > Table 3: True ground formulae of F(x,y):=R(x, y) -> S(y) w.r.t. true ground atoms R=((a1, b1), (a3, b2)) and S=((b1)) of p69 in [1]. X=(a1, a2, a3), Y=(b1, b2).
> > >
> > > |x | y |
> > > |---|---|
> > > |a1 | b1 |
> > > |a1 | b2 |
> > > |a2 | b1 |
> > > |a2 | b2 |
> > > |a3 | b1 |
> > >
> > >
> > > [1] https://web.cs.ucla.edu/~guyvdb/talks/IJCAI16-tutorial/IJCAI16-tutorial.pdf
> > > [2] https://simons.berkeley.edu/sites/default/files/docs/5662/talk-simons-2016.pdf

---

> > > > ### Comment · Reviewer_sKHS · 2022-12-08
> > > > **Question**
> > > >
> > > > Sorry for the late reply.
> > > > I don't think I understand the claim that "Einsum of LogicMP in MLN is counting all groundings in a single world". Can author clarify the scope of the latent variables v in Equation~1 for the smoker and friend example, if we have only two constants A and B. I initially thought the scope of the v covers all possible worlds, but it seems to be contradicting to the statement that LogicMP is counting all groundings in a single world.

---

> > > > > ### Author Response · Authors · 2022-12-09
> > > > > **LogicMP "counts" all groundings in a single world by variational marginal probabilities of ground atoms to sidestep counting all worlds**
> > > > >
> > > > > Thank you very much for your reply!
> > > > >
> > > > > The $\mathbf{v}$ is a set covering the unobserved ground atoms, e.g., $s(A), s(B)$. The range {$0, 1$}$^{|\mathbf{v}|}$ of $\mathbf{v}$ covers all possible worlds. In our variational algorithm, LogicMP mitigates the problem of counting exponential worlds by updating in the single world of approximate marginal probabilities of ground atoms where einsum is used to "count" all possible groundings.
> > > > >
> > > > >
> > > > > Let's take Fig. 1 in the current paper as an example to clarify the scope of $\mathbf{v}$ in Eq. 1. The predicates are smoke(x) ($s(x)$), friend(x, y) ($f(x, y)$) and cancer(x) ($c(x)$).  The obervations $O$ are two facts, i.e., $f(B,A)=true$ and $c(B)=true$.
> > > > > The unobserved variables $\mathbf{v}$ are {$v_{s(A)}, v_{s(B)}, v_{f(A,A)}, v_{f(A,B)}, v_{f(B,B)}, v_{c(A)}$}. Each variable $v_i$, e.g., $v_{s(A)} \in${$0, 1$}, is a binary discrete variable.
> > > > >
> > > > > An assignment to these variables corresponds to a world. A possible world is
> > > > >
> > > > > | |$v_{s(A)}$|$v_{s(B)}$|$v_{f(A,A)}$|$v_{f(A,B)}$|$v_{f(B,B)}$|$v_{c(A)}$|
> > > > > |---|---|---|---|---|---|---|
> > > > > |binary value|1|1|0|1|0|1|
> > > > >
> > > > >
> > > > > In our variational mean-field algorithm, each variable $v_i$ is represented by a marginal probability $Q_i(v_i) \in [0, 1]$, e.g., $Q_{s(A)}(1)=0.92(Q_{s(A)}(0)=0.08)$ denotes that the marginal of probability of $s(A)$ being true is 0.92 (0.08 for false). The marginals may be:
> > > > >
> > > > > | |$Q_{s(A)}(1)$|$Q_{s(B)}(1)$|$Q_{f(A,A)}(1)$|$Q_{f(A,B)}(1)$|$Q_{f(B,B)}(1)$|$Q_{c(A)}(1)$|
> > > > > |---|---|---|---|---|---|---|
> > > > > |continuous value|0.92|0.97|0.13|0.95|0.03|0.99|
> > > > >
> > > > > The mean-field update is computed by a "world" of these 6 marginals via Proposition 4, which updates new marginals of variables conditioned on the current marginals of variables. Note that when the marginals $Q_i(v_i)$ become discrete 0/1, the mean-field algorithm degenerates to the sampling algorithm (e.g., Gibbs sampling) that estimates the probability of each variable conditioned on the current discrete world and performs sampling (cf. Eq. 26 [1]). The difference is that the sampling-based MLN algorithm counts the groundings based on a discrete world while our mean-field algorithm "counts" the groundings based on the estimated marginals, e.g., the premise  $s(A) \wedge f(A, B)$ holds when $Q_{s(A)}(1)Q_{f(A, B)}(1)\approx 1$. This is why the einsum in LogicMP can be intuitively seen as counting groundings over a single world with marginal probabilities of the ground atoms.
> > > > >
> > > > > We illustrate how our algorithm updates the marginals of $s(A)$ and $s(B)$, i.e., $\mathbf{Q}_s$, as follows.
> > > > > They receive messages from 4 implications statements and we calculate them via einsum as illustrated in Fig. 1:
> > > > >
> > > > > |implication statement| einsum| variational approximation|
> > > > > |---|---|---|
> > > > > |$s(x) \wedge f(x, y) \to s(y)$| $e_1=$einsum('x, xy->y", $\mathbf{Q}_s(1)$, $\mathbf{Q}_f(1)$) |“counts” the true ground premises $s(x) \wedge f(x, y)$|
> > > > > |$\neg s(y) \wedge f(x, y) \to \neg s(x)$| $e_2=$einsum('y, xy->x", $1-\mathbf{Q}_s(1)$, $\mathbf{Q}_f(1)$)| “counts” the true ground premises $\neg s(y) \wedge f(x, y)$|
> > > > > |$c(x) \to s(x)$| $e_3=$einsum('x->x", $\mathbf{Q}_c(1)$)|“counts” the true ground premises $c(x)$|
> > > > > |$\neg c(x) \to \neg s(x)$| $e_4=$einsum('x->x", $1-\mathbf{Q}_c(1)$)|“counts” the true ground premises $\neg c(x)$|
> > > > >
> > > > > The updated marginals are:
> > > > >
> > > > > - $\mathbf{Q}_s(1) \sim \exp(\Phi_s(1) + w_1e_1 + w_2e_3)$
> > > > > - $\mathbf{Q}_s(0) \sim \exp(\Phi_s(0) + w_1e_2 + w_2e_4)$
> > > > >
> > > > > , where all computations are based on the "single world" with marginal probabilities of ground atoms, i.e, $Q_i$s.
> > > > >
> > > > > [1] https://www.cs.princeton.edu/courses/archive/fall11/cos597C/lectures/variational-inference-i.pdf

---

> > > > > > ### Comment · Reviewer_sKHS · 2022-12-13
> > > > > > **Regarding to the counting problem**
> > > > > >
> > > > > > Thank you for the clear explanation! It is supper appreciated!
> > > > > >
> > > > > > After discussing with other reviewers, we think the solution proposed here is innovating, but the current evaluations are mixing multiple contributions from the paper, which is difficult to understand the impact from each component.
> > > > > >
> > > > > > From the perspective of an scalable approximate inference algorithm on MLNs, author could compare with other MLNs approximate algorithms. The one proposed in Table~2 are good ones, but they are evaluated using the AUC metrics, which focuses too much on the perspective of learning representation instead of the inference quality. It would be nice if author could design some experiments to establish the efficiency of the proposed inference algorithm. For example, one can report the KL divergence between the exact marginal probability for each predicates, either by enumerating over all possible worlds or through exact lifted inference, v.s. the approximated marginal probability for each predicates using various approximated inference algorithm. Author could demonstrates the computation time on large network to demonstrate the scalability or compare the quality of the approximation given a limited time budget to demonstrate the quality of the estimate.
> > > > > >
> > > > > > From the learning representation perspective, author innovatively demonstrates one way to combine neural network backbone with some symbolic reasoning module on top. However, from the offline reviewer discussion, we agreed that author should establish a more comprehensive study as there are other established neural symbolic approaches that combines relational knowledge with a neural backbone, e.g. DeepProblog [1]. It would be nice author could demonstrate the representation of LogicMP vs. DeepProblog.
> > > > > >
> > > > > >
> > > > > > [1] - Manhaeve, Robin, et al. "Deepproblog: Neural probabilistic logic porgramming" Advances in Neural Information Processing Systems 31(2018)

---

### Author Response · Authors · 2022-11-16
**Response to all reviewers**

We sincerely thank all reviewers for your helpful suggestions and comments. The paper has been revised accordingly and the important modifications are highlighted in red color. To present within 9 pages after the modification, we temporarily move the "symbolic reasoning" paragraph in the related work to Appendix P. The main modifications include:

- **More details of the mean-field update**: we add more details in the derivation of the mean-field update equation in both the main paper (Sec. 3.1) and Appendix A.
- **More informative figure and examples**: we give an example of the calculation of the grounding message in the framework figure and describe it in Sec 4.1 to bridge the gap between the MLN and LogicMP.
- **More clear scope of the proposed method**: we emphasize that LogicMP is for the MLN inference where the MLN is fixed with no structure learning.
- **Fewer math notations**: we simplify the notations by removing unnecessary symbols when the context is clean.
- **Fewer vague sentences**: we revise or remove some vague statements to make them clear and precise.

We would greatly appreciate it if the reviewers can give us another pass on the paper.  Many thanks!

---

### Decision · Program_Chairs · 2023-01-20

**Decision:**

Reject

**Justification For Why Not Higher Score:**

We debated about the possibility of acceptance and we agreed that the current state of the paper would need some non-trivial reworking and some more experiments.

**Justification For Why Not Lower Score:**

N/A

**Metareview: Summary, Strengths And Weaknesses:**

The authors tackle the problem of performing inference in Markov logic networks (MNLs) and propose to speed up classical mean field (MF) approximations by introducing two observations, the ability to rule-out some groundings while computing messages in MF and the possibility to optimize the summations in the messages by rewriting these summations as a series of matrix multiplications (expressed in einsum notation). At the same time, they propose a neural backbone to be integrated in the inference process, realizing a neuro-symbolic hybrid approach via posterior regularization. This should put the proposed LogicMP on the same ball park of DeepProblog [1].

The reviewers appreciated the direction of the paper and the implementation perspective taken by the authors to speed up the grounding and inference in MLNs. This can have great potential. Several concerns were raised during the reviewing phase, however. These include three major factors: the presentation of the work, the empirical comparison and the theoretical speed up.

First, the proposed contribution sits in between proposing a neuro-symbolic approach but also a speed-up for classical MLN inference schemes. The way it is presented highlights mostly the second part, relegating the first in the Appendix, where however not enough details are provided. By not properly disentangle the two aspects, it is not clear which contributes most to the results and why people should adopt LogicMP.

In fact, more crucially, the experimental setting mostly concentrates on comparing the downstream task (classification accuracy) of approach on some classical MLN benchmark in a mixed setting that makes it somehow hard to clearly focus on evaluating the contribution along the two aforementioned dimensions: how good/relevant is the speed up w.r.t. other approximate MLN inference schemes and how does it compare against other neuro-symbolic reasoners (e.g., DeepProblog).
Specifically, the authors compare the gains in time only in Fig 6 measuring AUC-PR/minutes and number of groundings/seconds. But they do so on a subset of the datasets and not report raw times. It is important also to compare the optimized einsum selection w.r.t. a normal einsum operation. Furthermore, it is not clear if competitors can parallelize over the GPU as well.
Regarding competitors, the only neuro-symbolic competitor seems ExpressGNN, but it is orders of magnitude slower by design.

Lastly, the computational complexity gains are written in a misleading way. The first speed up from discarding non-relevant grounding looks sound but not surprising. Concerning the "einsum trick", The gain, as presented, seem to go from exponential to polynomial, but we should remark that operating in einsum notation cannot lower the computational complexity of an operation per se, but at most speed it up by not materializing in memory some intermediate products. Furthermore their exponential complexity is quite an extreme worst case bound that considers the most naive way to perform matrix (or tensor) multiplications. The gain in complexity that the authors highlight is due from using an optimized scheduler that breaks a chain of matrix multiplications into intermediate steps as to bound the complexity of each step to be worst-case cubic. Note that this can always be done, at the cost of finding the optimal breakdown. The cost of this optimization step is not discuss and does not seem to be easily bound. Worst-case, the optimizer might return the unoptimized chain of multiplications. This is why it is important to measure how a call to an unoptimized einsum operation can perform w.r.t. the optimized variant.

During the rebuttal, the authors tried to address all concerns and greatly improved the current status of the paper. Reviewers appreciated the effort. However, the discussion post rebuttal highlighted that some core concerns are still not properly addressed and opened up other perspective (discussed above) where the paper contribution is not clearly disentangled and properly evaluated. The paper would greatly benefit from incorporating the following suggestions
- Rewriting the presentation as positioning the paper in the broader neuro-symbolic literature and discussing why competitors such as
- Benchmarking inference quality (in terms of distance from true marginals/conditionals/likelihood) against approximate MLN schemes, after fixing the same MLN and weights and removing the neural backbone in LogicMP
- Compare against Lifted inference in MLN (it is not clear if BP/LiftedBP is the best of both or one of the two)
- Time comparison in seconds of the different ablation tests and approximate inference competitors on synthetic and real-world datasets while using the same CPU (or GPU) setting for all competitors.

The paper is rejected.

[1] - Manhaeve, Robin, et al. "Deepproblog: Neural probabilistic logic programming." Advances in Neural Information Processing Systems 31 (2018).

**Summary Of Ac-Reviewer Meeting:**

Reviewers agreed that the paper has some merits, but were still puzzled by many missing details.
The discussion post rebuttal helped highlighting that some core concerns are still not properly addressed and opened up other perspective (discussed above) where the paper contribution is not clearly disentangled and properly evaluated.

We finally agreed that the paper would greatly benefit from incorporating the following suggestions
- Rewriting the presentation as positioning the paper in the broader neuro-symbolic literature and discussing why competitors such as
- Benchmarking inference quality (in terms of distance from true marginals/conditionals/likelihood) against approximate MLN schemes, after fixing the same MLN and weights and removing the neural backbone in LogicMP
- Compare against Lifted inference in MLN (it is not clear if BP/LiftedBP is the best of both or one of the two)
- Time comparison in seconds of the different ablation tests and approximate inference competitors on synthetic and real-world datasets while using the same CPU (or GPU) setting for all competitors.